# Constrained Sampling for Language Models Should Be Easy: An MCMC Perspective

**Emmanuel Anaya Gonzalez**[1]* **Sairam Vaidya**[1]* **Kanghee Park**[1] **Ruyi Ji**[2]
**Taylor Berg-Kirkpatrick**[1] **Loris D'Antoni**[1]
[1]UCSD [2] Peking University
{fanayagonzalez,smahadevaganapathy,kap022,tberg,ldantoni}@ucsd.edu
jiruyi910387714@pku.edu.cn

## Abstract

Constrained decoding enables Language Models (LMs) to produce samples that provably satisfy hard constraints. However, existing constrained-decoding approaches often distort the underlying model distribution, a limitation that is especially problematic in applications like program fuzzing, where one wants to generate *diverse* and *valid* program inputs for testing purposes. We propose a new constrained sampling framework based on Markov Chain Monte Carlo (MCMC) that simultaneously satisfies three core desiderata: *constraint satisfying* (every sample satisfies the constraint), *monotonically converging* (the sampling process converges to the true conditional distribution), and *efficient* (high-quality samples emerge in few steps). Our method constructs a proposal distribution over valid outputs and applies a Metropolis-Hastings acceptance criterion based on the LM's likelihood, ensuring principled and efficient exploration of the constrained space. Empirically, our sampler outperforms existing methods on both synthetic benchmarks and real-world program fuzzing tasks [1].

## 1 Introduction

Language Models (LMs) have revolutionized a wide range of domains, from code generation [12] to automated reasoning [59]. Yet, ensuring that their outputs satisfy hard structural constraints—such as syntactic validity in domain-specific languages—remains a significant challenge [18]. While constrained decoding methods [9, 13, 19, 45, 50, 52, 53, 55, 44, 2, 36, 57] can enforce these constraints, they often distort the underlying generative distribution learned by the model, degrading performance in downstream tasks [54, 43].

This tradeoff is particularly detrimental in applications that rely not on a single high-quality sample, but on *diverse* samples from the constrained distribution. A compelling example is *program fuzzing*, a technique for discovering software bugs by automatically generating test inputs that explore different execution paths in a program. Modern fuzzers bootstrap this process using a small set of seed inputs, and the effectiveness of these seeds hinges on both their correctness—the seeds should be syntactically valid inputs that the program will not reject—and their distributional diversity—the seeds should exercise different execution paths. LMs offer a powerful mechanism to generate such seeds—but only if we can sample efficiently and faithfully from the constrained distribution.

Can we design *constrained sampling algorithms* that when executed for a given amount of time $t$ are *(i) Constraint satisfying:* produce a sample within $t$ time and that sample satisfies the constraint; *(ii)*

---

*Equal contribution.

[1]Code available at https://github.com/large-loris-models/casa

*Monotonically Converging:* converge to the true conditional distribution when $t$ goes to infinity; and *(iii) Efficient:* produce good estimates of the constrained distribution for low values of $t$?

We answer this question in the affirmative. Focusing on constraints expressed as *context-free grammars*, we introduce a family of sampling algorithms rooted in Markov Chain Monte Carlo (MCMC) techniques. Our key insight is to construct proposal distributions that generate only constraint-satisfying samples and use a Metropolis-Hastings criterion guided by the LM's likelihood function to accept/reject candidates. Unlike rejection sampling, every candidate in our method is valid by construction—ensuring *constraint satisfaction*. Our use of likelihood-aware MCMC transitions ensures *monotonic convergence*, and crucially, our empirical results reveal that the resulting chains converge *efficiently* to the desired conditional distribution.

We make the following contributions. First, we formalize the *desiderata* for constrained sampling—constraint satisfying, monotonically converging, and efficient—and show that existing methods fail to satisfy all three (Section 2). Second, we introduce an effective *MCMC-based framework* for constrained sampling that satisfies all three desiderata and instantiate it with three concrete proposal distributions (Section 3). Third, we validate our approach on both synthetic distributions and real-world fuzzing targets such as libxml2 and SQLite. The samples produced from our approach exhibit better KL divergence from the target distribution than competing approaches. Most importantly, program fuzzers (i.e., automated random testers) seeded with our samples consistently achieve higher code coverage on real-world tasks compared to fuzzers seeded using existing approaches (Section 4).

## 2 The Ideal Properties of Constrained Sampling

In this section, we formalize the problem of sampling from a language model (LM) conditioned on a constraint (i.e., constrained sampling), define our key desiderata of a good constrained sampling algorithm, and describe how existing constrained sampling algorithms do not meet such desiderata. We follow the definitions proposed by Park et al. [43].

**Language Models.** An (autoregressive) *language model* defines a probability distribution $P$ over sequences of tokens (i.e., sentences) $w \in \mathcal{V}^*$, where $\mathcal{V}$ denotes the vocabulary. The probability of a sequence is computed as the product of conditional probabilities for each token in the sequence:

$$P(w_1 \ldots w_n) = \Pi_{i=1}^n P(w_i \mid w_{1:i-1})$$

**Constraints and Grammars.** Given a LM $P$ and a constraint $\varphi$, the goal of constrained sampling is to sample sequences that satisfy the constraint. Constraints are typically specified as a language, which could be a regular language, context-free grammar (CFG), or other types of logical conditions that sequences must fulfill. In this work, we focus on constraints that can be defined by a context-free grammar, an expressive and formal way to define sets of valid sequences. For example, context-free grammars can express what programs in a given programming language are syntactically valid and what format a JSON object should abide to when being used to transfer data. While we focus on grammars, the techniques we propose can be applied to other types of constraints.

Formally, a context-free grammar $\mathcal{G} = (\Sigma, \mathcal{N}, S, \mathcal{R})$ consists of a set of terminal symbols $\Sigma$, a set of non-terminal symbols $\mathcal{N}$, a start nonterminal symbol $S \in \mathcal{N}$, and a set of production rules $\mathcal{R}$. A sequence $w$ is a valid sentence belonging to the language $\mathcal{L}(\mathcal{G})$ if it is derivable from the start symbol $S$ by applying a sequence of rules from $\mathcal{R}$. Each step in this derivation transforms a string $\alpha A \gamma$ into $\alpha \beta \gamma$ using a rule $A \rightarrow \beta \in \mathcal{R}$, and denoted by $\alpha A \gamma \Rightarrow \alpha \beta \gamma$.

**Example 1** (SQLite Test-Script Grammar). *Figure 1 illustrates a typical use-case: we ask a language model to generate SQLite regression test files (`.test`) that drives the SQLite engine down as many distinct execution paths as possible (Figure 1a). To exercise a specific component of the database, each file must satisfy the syntactic and semantic restrictions encoded in the SQLite test-script grammar shown in Figure 1b—e.g., the file should include a mandatory timeout directive `set ::timeout 60000` in the header and well-formed `do_execsql_test` blocks that wrap one or more SQL statements in braces and specify an expected result.*

*A complete derivation therefore starts from the non-terminal `root`, transforms (i.e., using $\Rightarrow$) into `header test_block_list finish`, and continues recursively until only quoted terminals remain. (See Appendix E.2 for a condensed parse tree.)*

```
Question 1: Generate a complete and complex sqlite          root ::=
.test file using different Tcl commands. Do not reuse           header test_block_list finish
previous solutions. Ensure you write a single .test        header ::=
file without any comments that ends with a "finish_test".       "set testdir [file dirname $argv0]"
                                                                "source $testdir/tester.tcl"
Solution 1:                                                     "set testprefix" identifier
set testdir [file dirname $argv0]                               "set ::timeout 60000"
source $testdir/tester.tcl                                      (variable_definition)*
...                                                         ...
ifcapable journalMode {                                    test_block_list  ::= (test_block)+
  catchsql {PRAGMA journal_mode=WAL}                       test_block ::= do_test | do_execsql_test
} {0 {}}                                                        | ifcapable_block | procedure_definition
finish_test                                                    | conditional_block | tcl_statement
                                                           do_execsql_test ::=
Question 2: Generate a complete and complex sqlite             "do_execsql_test" test_name
.test file using different Tcl commands. Do not reuse          "" sql_statements "" expected_result
previous solutions. Ensure you write a single .test        ...
file without any comments that ends with a "finish_test".  sql_statements ::=
                                                               sql_statement  (";" sql_statement)* ";"?
Solution 2:                                                finish ::= "finish_test"
```

|                (a) Prompt                |                (b) Grammar                |

Figure 1: (a) Prompt to generate seed test cases for fuzzing the SQLite engine. (b) Simplified version of the SQLite test-script grammar written in EBNF notation. The goal of the problem is to generate multiple diverse seeds that trigger different code paths in the library being tested.

**Grammar-Aligned Sampling.** Grammar-aligned sampling aims to sample sequences from $P$ that belong to the language $\mathcal{L}(\mathcal{G})$, while preserving the model's underlying distribution. This can be viewed as sampling from the constrained distribution $P^{\mathcal{G}}$, which is proportional to the original model distribution but restricted to sequences that satisfy the constraint. Mathematically, for a given grammar $\mathcal{G}$ and model $P$, we want to sample from

$$P^{\mathcal{G}}(w) = \frac{\mathbb{1}[w \in \mathcal{L}(\mathcal{G})] \cdot P(w)}{\sum_{w'} \mathbb{1}[w' \in \mathcal{L}(\mathcal{G})] \cdot P(w')}$$

## 2.1 Limitations of Existing Approaches

**Rejection Sampling.** A common method for obtaining constrained samples from a language model is *rejection sampling*, which repeatedly draws outputs from the model and discards ones that do not satisfy the constraint. While samples accepted via this process correctly follow the language model's distribution conditioned on the constraint, there is no guarantee on how many samples one needs to reject before getting a sample satisfying the constraint. This inefficiency becomes especially pronounced when the constraint describes a pattern that is infrequent or not naturally favored by the model's learned distribution, often requiring many rejections before a valid sample is drawn. For example, for the problem in Figure 1, out of 500 samples produced by Llama-3.1-8B-Instruct, only 2 (0.4%) satisfied the constraint imposed by the grammar.

**Constrained Decoding.** The inefficiency of rejection sampling has led to *constrained decoding algorithms* [50, 9, 19] that at each decoding step evaluate the LM next tokens against the specified constraints. Invalid tokens that cause the generated sequence to not satisfy the given constraint are masked from the probability distribution, forcing the model to select tokens that will lead to *constraint-satisfying* sequences. In particular, when the constraint is a context-free grammar, the technique is known as Grammar-Constrained Decoding (GCD) [19].

As shown by Park et al. [43], constrained decoding *does not preserve the underlying distribution of the model*. If we define the prefix language $\mathcal{L}_{\text{prefix}}(\mathcal{G}) = \{w \in \Sigma^* \mid wv \in \mathcal{L}(\mathcal{G})\}$ of a grammar $\mathcal{G}$ as the set containing all possible prefixes of sentences in the grammar's language, the distribution captured by GCD is the following incorrect conditional distribution:

$$\tilde{P}^{\mathcal{G}}_{\text{GCD}}(w_i \mid w_{1:i-1}) = \frac{P(w_i \mid w_{1:i-1}) \cdot \mathbb{1}[w_{1:i} \in \mathcal{L}_{\text{prefix}}(\mathcal{G})]}{\sum_{w'_i} P(w'_i \mid w_{1:i-1}) \cdot \mathbb{1}[w_{1:i-1}, w'_i \in \mathcal{L}_{\text{prefix}}(\mathcal{G})]}$$

For example, for the problem in Figure 1, regardless of how many samples one generates using GCD with Llama-3.1-8B-Instruct the empirical KL divergence from the sample distribution to $P^{\mathcal{G}}$ does not decrease.

If we assume that the LM is good at sampling good seeds for a fuzzer, this unfaithfulness to the target distribution results in worse samples that cover fewer code paths (as shown in Section 4.2).

**Adaptive Sampling with Approximate Expected Futures (ASAp).** Park et al. [43] showed how to correct next-token conditional distribution for grammar-aligned sampling using the notion of *Expected Future Grammaticality* (EFG), defined as $c(w_{1:i}) = \mathbb{E}_{P(w_{i+1:n}|w_{1:i})}[\mathbb{1}[w \in \mathcal{L}(\mathcal{G})]]$—i.e., the probability that sampling a continuation of the prefix $w_{1:i}$ will lead to a valid sequence in the grammar. The conditional probability required by grammar-aligned sampling can be then written as:

$$P^{\mathcal{G}}(w_i \mid w_{1:i-1}) = \frac{P(w_i \mid w_{1:i-1}) \cdot c(w_{1:i})}{\sum_{w'_i} P(w'_i \mid w_{1:i-1}) \cdot c(w_{1:i-1}, w'_i)} \tag{1}$$

Park et al. [43] proposed Adaptive Sampling with Approximate expected futures (ASAp) to approximate grammar-aligned sampling. ASAp iteratively overapproximates expected future grammaticality by removing probability mass associated with invalid prefixes identified from previous samples. While in the limit this approach reaches the desired distribution, it does not do so *monotonically*—i.e., it can produce intermediate EFG approximations that are very skewed. Park et al. [43] empirically showed that it can take thousands of samples for ASAp to start converging, making the algorithm practically *not efficient*. For example, for the problem in Figure 1, 100 samples generated using ASAp with `Llama-3.1-8B-Instruct` exhibited worse empirical KL divergence from $P^{\mathcal{G}}$ than even GCD!

## 2.2 Desired Properties of a Grammar-Aligned Sampler

We formulate properties we claim a good constrained-sampling algorithm should satisfy. Because convergence in the limit is impractical, we assume that an algorithm $S$ is given a finite amount of time $t$ to produce a sample, and we want the algorithm to satisfy three properties:

**Constraint Satisfying:** $S$ produces a sample within $t$ time and that sample satisfies the constraint;

**Monotonically Converging:** the total variance distance between the output distribution of $S$ and the target distribution $P^{\mathcal{G}}$ monotonically decreases and converges to $0$ as $t$ approaches infinity;

**Efficient:** $S$ produces good estimates of the constrained distribution for low values of $t$.

By focusing on these properties, we aim to provide a sampling approach that balances efficiency, reliability, and correctness, addressing the shortcomings of existing methods, which fail to achieve all three goals simultaneously. In Section 3, we present our MCMC-based approach for constrained decoding, which provably guarantees the first two properties and is in practice efficient (Section 4). During the development of this work, concurrent research [38, 37] has introduced sequential Monte Carlo algorithms that also align with our desiderata. We empirically evaluate their performance in Section 4 and discuss how they relate to our approach in Section 5.

## 3 Grammar-Aligned MCMC Sampling

We propose a constrained sampling framework based on Markov Chain Monte Carlo (MCMC) that operates strictly within the space of grammar-valid sequences. Rather than relying on rejection of ungrammatical outputs, our method uses grammar-constrained decoding (GCD) to iteratively refine samples through local proposals that are always constraint-satisfying—a property guaranteed by GCD. This framework provides a principled mechanism to balance computational cost and sampling fidelity: as the chain progresses, it converges toward the true grammar-aligned distribution. Intuitively, our approach uses MCMC to turn any GCD implementation into a grammar-aligned sampler.

### 3.1 Constrained Generation via Metropolis-Hastings

We use the Metropolis-Hastings algorithm [23], a standard MCMC method, to construct a Markov chain whose stationary distribution matches a desired target $\pi$ over a set of states $S$. Given a *proposal distribution* $q(y \mid x)$, which defines how to sample a candidate state $y$ from a current state $x$, the algorithm accepts the proposed candidate with probability $\alpha(x, y)$, defined as:

$$\alpha(x, y) = \min\left\{1, \frac{\pi(y)q(x \mid y)}{\pi(x)q(y \mid x)}\right\} \tag{2}$$

**Algorithm 1:** The Metropolis-Hastings algorithm instantiated for grammar-aligned sampling.

---

**Data:** the LM $P$, the grammar $\mathcal{G}$, a parameter $k$ denoting the chain length, and a configurable distribution $p_{\text{POS}}^w$ for sampling a random prefix from a given sequence.

**Result:** a proposed sequence $s'$.

| | |
|---|---|
| **1** $w_0 \leftarrow$ a GCD sample; | **8 Function** Propose($w$)**:** |
| **2 foreach** $i \in 1 \dots k$ **do** | **9** $\quad$ $i \leftarrow$ an index sampled from $p_{\text{POS}}^w$; |
| **3** $\quad$ $w \leftarrow$ Propose($w_{i-1}$); | **10** $\quad$ $w^p \leftarrow w_{1:i}$; |
| **4** $\quad$ $a \leftarrow \alpha(w_{i-1}, w)$; | **11** $\quad$ **return** a GCD sample with the prefix $w^p$; |
| **5** $\quad$ $b \leftarrow$ Bernoulli($a$); | |
| **6** $\quad$ **if** $b$ **then** $w_i \leftarrow w$ **else** $w_i \leftarrow w_{i-1}$; | |
| **7 return** $w_k$; | |

---

This acceptance rule compares how likely $y$ is under the target distribution to how likely $x$ is, adjusted by the relative likelihood of proposing each direction, $q(x \mid y)$ and $q(y \mid x)$. Intuitively, proposals that improve the target probability are usually accepted, while worse ones are accepted with a controlled probability to maintain exploration. The acceptance rule ensures detailed balance—a property that guarantees the target distribution is stationary for the Markov chain—which in turn ensures that the chain will converge to the desired distribution over time.

Alg. 1 presents our instantiation of the Metropolis-Hastings algorithm for grammar-aligned sampling, where the target distribution is the constrained distribution $P^{\mathcal{G}}(w)$. The algorithm starts with a random GCD sample (Line 1)—which is guaranteed to be constraint-satisfying—and refines it toward the target distribution $P^{\mathcal{G}}(w)$ by running the constructed Markov chain for $k$ steps (Lines 2–7). In each step, the algorithm simulates the Markov chain by first drawing a new sample from the proposal distribution (Line 3) and then accepting it with probability $\alpha(w_{i-1}, w)$ (Lines 4–6). Finally, the last sample is returned as the result (Line 8).

One important point to note is the computation of the acceptance probability $\alpha(w_{i-1}, w')$ (Line 4), whose definition relies on the target probabilities $P^{\mathcal{G}}(w_{i-1})$ and $P^{\mathcal{G}}(w')$. However, these probabilities are difficult to compute in practice because their normalization factor $\sum_{w'} \mathbb{1}[w' \in \mathcal{L}(\mathcal{G})] \cdot P(w')$ requires summing over the whole language, which is typically intractably large. Fortunately, this factor cancels out from the acceptance probability after unfolding the target probabilities, as shown below, which induces an efficient implementation for the function $\alpha$:

$$\alpha(w, w') = \min\left\{1, \frac{P^{\mathcal{G}}(w')q(w \mid w')}{P^{\mathcal{G}}(w)q(w' \mid w)}\right\} = \min\left\{1, \frac{\mathbb{1}[w' \in \mathcal{L}(\mathcal{G})] \cdot P(w') \cdot q(w \mid w')}{\mathbb{1}[w \in \mathcal{L}(\mathcal{G})] \cdot P(w) \cdot q(w' \mid w)}\right\}$$

The last component in Alg. 1 is the proposal distribution (Line 3). The Metropolis-Hastings algorithm offers considerable flexibility in this distribution — any choice satisfying some mild conditions (i.e., irreducibility and aperiodicity) can yield a theoretically sound sampler. Hence, we propose a parameterized family of proposal distributions for grammar-aligned sampling to fit different scenarios (Lines 9–12), which we elaborate on in Section 3.2.

## 3.2 Proposal Distributions for Grammar-Aligned MCMC

The performance of MCMC for grammar-aligned sampling is tied to the proposal distribution. Different proposals make different trade-offs: some are simple to implement but mix slowly, while others complex ones may offer faster convergence. There is no single best choice across all settings.

In this section, we present a parameterized family of proposal distributions that is *(i)* tailored to sequences generated by a language model, and *(ii)* maintains constraint satisfaction by construction. As shown in Lines 9–12 of Alg. 1, the proposal mechanism operates by selecting a prefix of the current sequence and resampling the remainder using GCD. Given a sequence $w$ and a distribution $p_{\text{POS}}^w$ over the token positions $[0, |w|]$, we sample an index $i$ from $p_{\text{POS}}^w$ and extract the prefix $w_{1:i}$. A new candidate $w'$ is then generated by running a grammar-constrained decoder (GCD) conditioned on this prefix. This strategy allows local, structured edits while ensuring that every proposed sequence remains within the constrained language. For example, consider a grammar describing only sequences of 0s and 1s and let's assume GCD has produced the sequence 01001. Our MCMC algorithm can sample any prefix of this sequence, let's say 010, and produce from it a continuation using GCD, e.g.,

it could produce the sequence 01011111. Whether the new sequence will be accepted depends on the likelihood of the two sequences according to the LM.

Under this framework, we prove Alg. 1 exhibits the desired properties of a grammar-aligned sampler – satisfying both the *constraint satisfying* and *monotonically converging* properties (Sec. 2.2) if the truncation distribution $p_{\text{POS}}^w$ always assigns a non-zero probability to the empty prefix (Appendix F).

To conclude this section, we describe three concrete instantiations of this framework, each corresponding to a different choice of prefix distribution, $p_{\text{POS}}^w$.

**Uniform.** This proposal distribution samples a truncation point uniformly at random from the positions in the current sequence—i.e., $p_{\text{POS}}^w$ is the uniform distribution. This proposal creates local moves that preserve partial structure while allowing for diversity in continuations.

**Priority.** This proposal biases the truncation toward parts of the sequence where the model is uncertain or poorly aligned with the grammar, enabling targeted refinement of weaker regions. We use *perplexity*, a common measurement of uncertainty, to carry in this bias and set the truncation distribution $p_{\text{POS}}^w$ to the LM's token-level perplexity, i.e., $p_{\text{POS}}^w(i) \propto \text{PP}(P(\cdot \mid w_{1:i}))$. In this way, positions with higher uncertainty will have a greater probability to be resampled.

**Restart.** The simplest proposal always discards the current sample and generates a new one from scratch using GCD—$p_{\text{POS}}^w$ selects position 0 with probability 1. Since the proposal is independent of the current state, this reduces to resampled importance sampling [14].

## 4 Experiments

In this section, we show that MCMC yields better samples than existing approaches in a compute-matched setting. In Sec. 4.1 we demonstrate empirically that MCMC converges to $P^{\mathcal{G}}$ using fewer samples on constrained generation benchmarks proposed in Park et al. [43] and Lipkin et al. [37]. In Sec. 4.2 we show that MCMC, when compared to alternative sampling techniques, improves the quality of seeds needed to bootstrap fuzzing algorithms.

For the alternative sampling methods, we compare against vanilla GCD [19], ASAp [43], and SMC+AWRS [38, 37], a recently proposed constrained sampling algorithm that has goals similar to ours. SMC+AWRS performs Sequential Monte Carlo on $n$ particles (sequences), each of which is extended using token-level adaptive rejection sampling; we compare this setting to running MCMC for $n$ steps. In addition, we also evaluated the performance of rejection sampling in all of these domains, but found it exhibits impractically low acceptance rates (<1%), so we do not report it as a baseline in our evaluation. The main text presents results for `Llama-3.1-8B-Instruct`; additional models are evaluated in Appendix D and Appendix E.10. We implemented our MCMC framework as an extension of the Transformers-GAD library [43].

### 4.1 Domain-Specific Constrained Generation

We first evaluate the different sampling methods on four grammar-constrained generation tasks proposed by Park et al. [43] and Lipkin et al. [37]. From [43], two of our tasks involve synthesizing expressions in an extension of linear integer arithmetic (SLIA) and loop invariants with bit-vector arithmetic (BV4). The problems are expressed as Syntax-Guided Synthesis problems (SyGuS) [4], a standardized format where a logical specification and a context-free grammar of first-order terms are provided and the goal is to synthesize a term in the grammar that satisfies the specification. The prompts consist of 3 in-context examples of the form [specification, solution] and the grammar is then provided as a constraint for grammar-aligned sampling. Our third task is the constituency parsing (CP) task already used in prior GCD work [19] where the grammar is used to help the model produce well-parenthesized parse trees for English sentences. For our fourth domain, we draw the Molecular Synthesis (MS) task from [37], where the objective is to generate drug-like compounds in the SMILES format from in-distribution examples. In total, our evaluation set contains 15 SLIA problems, 14 BV4 problems, 6 CP problems and 1 MS problem.

We run all sampling techniques and compare them in a compute-matched setting, each with a budget of $n$ generated sequences for sample, for $n \in [1, \ldots, 10]$. We write MCMC-T ($k = n$) to denote the final sample after running MCMC with proposal distribution T$\in$ {Uniform, Priority, Restart} for $n$ steps. Similarly, we write ASAp ($k = n$) to denote a sample from the distribution learned

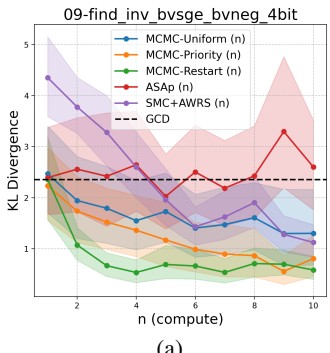
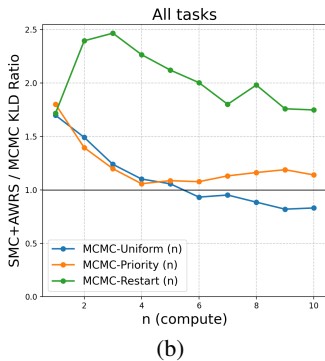

|   (a)   |   (b)   |

Figure 2: (a) KL divergence of different sampling methods, on a representative SYGUS task. (b) Geomean ratio between KL divergence of MCMC vs SMC, across all tasks ($> 1$ is better).

after having $n$ observed samples. Finally, SMC+AWRS ($k = n$) denotes a Particle Filter run with $n$ particles, and a final resample step to obtain a single, final sequence.

**Measures.** An ideal measure to compare sampling approaches is the distance between the sample distribution and the target $P^{\mathcal{G}}$. It is, however, impractical because we can neither exhaust the infinite sequence domain nor evaluate any probabilities in $P^{\mathcal{G}}$. Therefore, we follow Park et al. [43] to use an approximate measure instead, which is the KL divergence to the LM distribution $P$ on the finite set of all observed samples. This approximation aligns well to the ideal measure because the LM's distribution $P$ is proportional to the target $P^{\mathcal{G}}$ on valid samples: $\mathrm{KL}(\tilde{P^{\mathcal{G}}} \| P) = \mathbb{E}_{\tilde{P^{\mathcal{G}}}} \left[ \log \frac{\tilde{P^{\mathcal{G}}}}{P} \right] = \mathbb{E}_{\tilde{P^{\mathcal{G}}}} \left[ \log \frac{\tilde{P^{\mathcal{G}}}}{C \cdot P^{\mathcal{G}}} \right] = \mathbb{E}_{\tilde{P^{\mathcal{G}}}} \left[ \log \frac{\tilde{P^{\mathcal{G}}}}{P^{\mathcal{G}}} \right] - \log C = \mathrm{KL}(\tilde{P^{\mathcal{G}}} \| P^{\mathcal{G}}) - \log C$, where $\tilde{P^{\mathcal{G}}}$ denotes the sample distribution, and $C$ is a constant.

For each individual task, MCMC variant, and number of steps, we obtain 100 samples and use bootstrapping [16] to report mean KL divergence and 95% confidence interval. We do analogously for ASAp and SMC+AWRS.

**Results and Findings.** Figure 2a illustrates how the KL divergence for our MCMC approaches monotonically decreases with the number of steps for one representative problem from the SYGUS benchmark (the other tasks show similar trends). The KL divergence decreases and converges in trend for all variants of MCMC, though with some fluctuation caused by randomness. Nonetheless, for any given number of steps, MCMC samples are always closer in distribution to $P^{\mathcal{G}}$ than GCD. All variants of MCMC display large reductions in KL divergence with respect to GCD even after a handful of steps (at 3 steps: geomean. $1.57\times$ reduction for Uniform, $1.79\times$ for Priority, and $3.10\times$ for Restart; at 10 steps: geomean. $2.11\times$ for Uniform, $2.42\times$ for Priority, and $5.07\times$ for Restart).

In contrast, although ASAp also converges to $P^{\mathcal{G}}$ in the limit, it does not exhibit monotonic convergence, and in practice it yields worse KL divergence than GCD in many cases. For this reasong, larger reductions are observed on average when comparing MCMC to ASAP for the same number of steps (at 3 steps: geomean. $1.86\times$ reduction for Uniform, $1.98\times$ for Priority, and $3.36\times$ for Restart; at 10 steps: geomean. $2.25\times$ reduction for Uniform, $3.53\times$ for Priority, and $5.42\times$ for Restart).

SMC+AWRS also displays convergence to the target distribution. Interestingly, it tends to perform worse than GCD with a small number of particles ($< 3$); this might be an artifact of the AWRS token-level sampler since, in theory, when using the standard token-masking approach, SMC ($k = 1$) is equivalent to GCD, as is the case for MCMC ($k = 1$) and ASAp ($k = 1$). All variants of MCMC display better performance than SMC+AWRS in a low-compute regime ($n < 5$), whereas the latter overtakes MCMC-Uniform when more generations are allowed. This trend is illustrated in Figure 2b. Points above $y = 1$ indicate that the corresponding MCMC variant results in lower KL divergence for the same number of sequences used. MCMC-Priority and MCMC-Restart outperform SMC+AWRS for the evaluated compute ($> 1.1\times$ and $> 1.7\times$, resp.).

Across all tasks, MCMC-Restart consistently exhibits better quality of approximation than the rest of the sampling methods for equivalent compute, including MCMC-Uniform and MCMC-Priority. This

is a notable result, since MCMC-Restart does not accumulate any information about previous states in the Markov Chain random walk.

## 4.2 Fuzzing experiments

Coverage-guided fuzzing [10] iteratively mutates seed inputs to randomly generate test cases that exercise as many execution paths as possible in a target binary (a practice that can often reveal bugs in the code under test). The quality of the initial seed corpus—particularly their structural validity and diversity—can significantly impact downstream coverage, i.e., how many execution paths the fuzzer can exercise [24]. In this section, we evaluate whether different techniques for grammar-constrained sampling can be used to produce high-quality seeds for a state-of-the-art program fuzzer, AFL++ [17]. In our experiment, the grammar is used to guarantee that the LM produces inputs for the library under test that are *valid* (they will not be immediately rejected by the program) and can trigger execution of specific components of a library (e.g, forcing the SQLite timeout directive).

**Benchmarks.**  We evaluate our approach on two widely adopted, grammar-intensive targets: XML (using the libxml2 parser [56]) and SQL (using the SQLite engine [25]). To reflect realistic use cases where a general user prompt remains fixed while grammar constraints evolve to target different code components, we employ one high-level prompt per library (e.g., the one in Figure 1a) and introduce domain-specific constraints that can trigger different code components directly into the grammar. Our grammars are specializations of publicly available EBNF grammars for XML and SQL [22, 21, 62]. For XML, we modify the grammar to enforce mandatory `<!ENTITY>` and `<!ELEMENT>` declarations within the DOCTYPE, while for SQL we mandate a `set ::timeout 60000` directive within each `.test` file. A snippet of the grammar we use to test the SQL engine is given in Figure 1b and the full list of targets, versions, and seed formats is given in Appendix E.1. A user of the fuzzer can modify these grammars to stress test different components of the software under test.

**Measures.**  Aside from the KL divergence, which we measure in the way discussed in Section 4.1, we want to measure whether a better sampler leads to the fuzzer having higher branch coverage—i.e., the number of unique executed code branches over the total number of branches in the software under test (computed via LLVM instrumentation [1]).

We evaluate our methods across two dimensions: seed quality and computational budget. First, to assess seed quality, we generate 100 seeds per method, using GCD, ASAp; SMC+AWRS, MCMC-Priority, MCMC-Restart, and MCMC-Uniform with varying number of steps ($k \in \{2, 5, 10\}$). Second, to evaluate the trade-off between seed quality and quantity, we vary the number of initial seeds ($N \in \{50, 100, 200, 500\}$) for compute-matched comparisons—since MCMC with $k = 10$ takes approximately $10\times$ longer per seed than GCD, we can compare how fuzzing coverage varies when using 50 MCMC seeds versus 500 GCD seeds generated in the same amount of time.

We also include as a baseline Grammarinator [26], a fuzzing tool for generating random strings in a grammar that does not use LMs. For these benchmarks, rejection sampling exhibits impractical acceptance rates (<1%) (Appendix E.6) and we therefore exclude it from our comparison as it would take more than 10,000 samples to produce 100 valid seeds.

We describe the full fuzzing protocol in Appendix E.4. In summary, for every benchmark and method, we generate $N \in \{50, 100, 200, 500\}$ seeds for AFL++ and run it for 6 hours (in this time AFL++ generates thousands of new inputs based on the seeds and executes the software on them). We then measure mean branch coverage at different time steps with bootstrapped 95% confidence intervals over five independent 6-hour-long fuzzing trials, following standard fuzzing-evaluation protocols [11, 31].

**Findings.**  We present representative results for the SQL benchmark using Llama-3.1-8B-Instruct (full results in Appendix E.10). Figure 3a illustrates how the KL divergence changes at different steps in a similar way as observed in Section 4.1 for synthetic benchmarks—it decreases and converges in trend for all variants of MCMC, though with some fluctuation caused by randomness.

The key result is given in Figure 3b. To avoid clutter,Figure 3b only reports the results for MCMC when run for 10 steps (the versions for 2 and 5 can be found in Appendix E.8 and E.10).

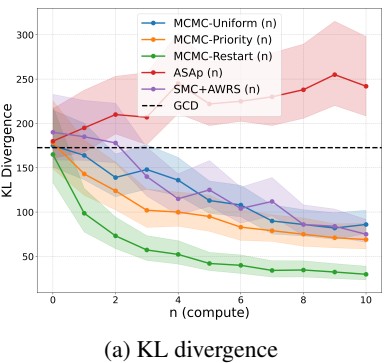

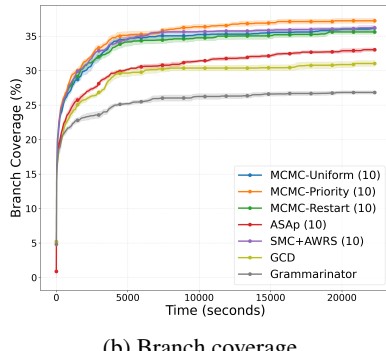

(a) KL divergence

(b) Branch coverage

Figure 3: SQL benchmark: (a) KL divergence for different sampling methods. (b) Branch coverage over time (100 seeds, 6-hour runs).

Fuzzing using seeds produced via MCMC and SMC+AWRS leads to significantly higher branch coverage than Grammarinator, GCD, and ASAp. Grammarinator, relying solely on grammar, achieves the lowest coverage, followed by GCD and then ASAp. Among all approaches, MCMC-Priority ($k = 10$) is the most effective, delivering branch coverages of 36.69% on SQL and 12.79% on XML—corresponding to gains of $1.17\times$ and $1.15\times$ over GCD (31.52%, 11.08%) and $1.12\times$ and $1.13\times$ over ASAp (32.61%, 11.30%) on the SQL and XML benchmarks, respectively. SMC+AWRS ($k = 10$) achieves competitive performance with 36.44% on SQL and 12.63% on XML, demonstrating $1.14\times$ and $1.13\times$ gains over GCD while remaining $1.04\times$ below to MCMC-Priority.

This result underscores the benefit of MCMC in diversifying the kind of samples an LM can produce. As expected the versions of MCMC with ($k = 5$) provide coverages that are lower than at ($k = 10$), but higher than GCD; at ($k = 2$) the coverage is quite similar to that of GCD.

Intriguingly, the MCMC methods coverage ranking (Figure 3b) is inversely mirrored by their KL divergence from the true conditional grammar-constrained distribution (Figure 3a), though the overall differences are relatively minor. This phenomenon suggests that while LMs provide a strong foundation for producing diverse outputs, approaches like MCMC-Priority may be better at exploring "variants" of the same output, which in turn are better for exercising branch coverage in fuzzing.

Since MCMC with $k = 10$ takes approximately $10\times$ longer than GCD per seed (Appendix E.7), we compared MCMC with 50 seeds against GCD with 500 seeds (compute-matched). Remarkably, MCMC-Priority with just 50 seeds achieved 34.48% coverage on SQL, surpassing GCD with 500 seeds at 31.70% (see Appendix E.8 for complete results across all models). This experiment demonstrates that sampling fidelity dominates sheer quantity for fuzzing effectiveness, likely because high-quality diverse seeds enable the fuzzer to explore more productive regions of the input space and avoid wasting time mutating low-quality seeds.

In summary, the results support the claim that MCMC-based constrained sampling monotonically converges to the desired distribution, and convergence is indicative of higher coverage in fuzzing.

## 5 Related Work

**Grammar-Aligned Sampling.** Several recent works have emphasized the importance of sampling multiple outputs from LMs to approximate the model's distribution [29, 48, 43]. However, when hard constraints are introduced, typical decoding strategies like top-$k$, nucleus, or beam search become unreliable estimators of the constrained distribution.

Park et al. [43] introduced the ASAp to approximate the true constrained distribution, which it has been studied further by Melcer et al. [42]. However, these methods are slow to converge, requiring thousands of samples, and therefore inefficient in practice.

Recent work [3, 37, 38] introduces alternative Monte Carlo and resampling algorithms for correcting the distortion problem induced by locally constrained decoding. Ahmed et al. [3] takes a three-step approach of first obtaining an unconstrained (and unbiased) sample, project it to the constrained space via a *pseudolikelihood* function, and finally apply resampled importance sampling to correct for bias.

Loula et al. [38] combine Sequential Monte Carlo and Importance Sampling to enforce not only local constrains, but also semantic constraints via *expensive potentials* that cannot be evaluated incrementally. Both SMC and MCMC are theoretically guaranteed to produce samples from the target distribution as the number of particles or steps, respectively, goes to infinity, but neither directly reduces to the other in the general case. From the implementation perspective, in SMC, the number of particles needs to be fixed beforehand and all must be kept in memory during execution, whereas MCMC is an anytime algorithm, and only a single, complete sequence is maintained at all times.

Lipkin et al. [37] presents a token-level adaptive rejection sampling algorithm (AWRS) which, combined with SMC sequence sampling, yields a strong baseline for constrained generation. The AWRS sampler can be integrated to other constrained generation algorithms including MCMC; we leave the details and further analysis as future work.

**Controlled Generation.** A large body of work has investigated constrained decoding methods that modify the token-by-token decoding process of LMs to enforce syntactic or lexical restrictions. These constraints are often specified using regular languages [41, 58] or context-free grammars (CFGs) [9, 13, 19, 45, 50, 52, 53, 55, 44, 2, 36, 57, 6, 27, 28, 39, 40, 46]. As discussed in Section 2.1 and as observed by Park et al. [43], many approaches that enforce constraints incrementally distort the LM distribution and do not converge to the target constrained distribution.

Gradient-based sampling methods [5, 32, 35, 47] offer a softer alternative, guiding generation toward constraint satisfaction by using relaxed, differentiable surrogates. These methods are better suited for soft or semantic constraints but still suffer from inefficiency and are not guaranteed to produce constraint-satisfying outputs.

Methods such as GeLaTo [60] and Ctrl-G [61] combine autoregressive language models with Hidden Markov Models (HMMs) to guide generation based on constraints. These techniques are specifically designed for constraints that can be represented as deterministic finite automata (DFA). Given a prefix, a pretrained HMM is used to approximate the probability that a subsequently generated suffix will satisfy the DFA constraint, effectively estimating the likelihood of a valid continuation. However, this approach is limited to DFA-representable constraints and cannot be easily extended to more general grammars like context-free grammar. Furthermore, these methods require training separate surrogate models. Crucially, they also do not guarantee convergence to the ideal distribution, a limitation they share with other techniques that use approximate inference for intractable conditional distributions (e.g., Feynman-Kac Transformer Models) [47, 34].

**Proposal Distributions.** Our work implements a vanilla Metropolis-Hastings sampler with proposal distributions that arise naturally from the autoregresive nature of language models. Related ideas from the literature on sampling form energy-based models [49, 33] could yield further improvements. Proposal distributions that leverage infilling [8], the constraining grammar, or learning [51] seem promising to incentivize exploration of more promising regions of sequence space. Further performance improvements are applicable to MH when the proposal distribution is independent from the current state [7], as is the case with our MCMC-Restart variant.

# 6 Conclusion

We introduced a simple yet effective MCMC-based framework for constrained decoding. Unlike prior approaches that suffer from slow convergence or rely on inefficient rejection sampling, our method directly samples from within the constrained space while asymptotically preserving the target distribution defined by the LM. Our framework leads to practical improvements in real-world applications where sampling diverse inputs is crucial—most notably, program fuzzing, where high-quality diverse samples translate into higher code coverage.

Our evaluation spans program synthesis benchmarks (SLIA, BV4, constituency parsing) and two grammar-intensive fuzzing targets (libxml2 and SQLite), each with one high-level prompt per task. While we find consistent trends across different language models and extended 6-hour fuzzing runs, broader evaluation across additional domains (e.g., protocol parsers, configuration languages) and longer campaign durations (24+ hours) would further strengthen generalizability claims. While we study three MCMC proposal distributions, our framework opens the door for exploring richer proposal mechanisms and integration with other decoding strategies.

## Acknowledgments

This work was supported in part by a Microsoft Faculty Fellowship; a UCSD JSOE Scholarship; Google's Gemma Academic Program GCP Credit Award; and NSF under grants CCF-2422214, CCF-2402833 and CCF-2211968. Any opinions, findings, and conclusions or recommendations expressed in this publication are those of the authors, and do not necessarily reflect the views of the sponsoring entities. Loris D'Antoni holds concurrent appointments as a Professor at the University of California San Diego and as an Amazon Scholar. This paper describes work performed at the University of California San Diego and is not associated with Amazon. The authors deeply thank Nadia Polikarpova for her insightful feedback and guidance during the development of this project.

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

# Appendix

## A    Hardware and Software

Our experiments were conducted on Ubuntu 22.04 LTS nodes with Intel Xeon Gold 6230 CPUs (2.10 GHz, 10 cores, 20 threads allocated) and 384 GB RAM. For GPU-accelerated workloads, we provisioned 2x NVIDIA RTX A6000 GPUs. Our implementation is based on Python 3.10.12, PyTorch 2.6.0+cu124, AFL++ 4.00c and LLVM 14.0.0.

## B    Hyperparameters

For language-model decoding, we set temperature to 1.0, top-p to 1.0, and top-k to 0 to allow sampling from the full token vocabulary. We limited the maximum number of newly generated tokens to 512 for XML, and 1024 for SQL `.test` scripts.

## C    Model Checkpoints

We evaluate on two instruction-tuned models representing different architectural families:

- **Llama-3.1-8B-Instruct** [20]: `https://huggingface.co/meta-llama/Llama-3.1-8B-Instruct` (commit `0e9e39f`)
- **Qwen2.5-Coder-7B-Instruct** [30]: `https://huggingface.co/Qwen/Qwen2.5-Coder-7B-Instruct` (commit `c03e6d3`)

All models use BF16 precision with their default tokenizers and system prompts.

## D    Domain-Specific Generation Benchmarks

We evaluate the convergence properties of Grammar-Aligned MCMC Sampling empirically on the benchmark tasks proposed by Park et al. [43]. Fig. 4 relates the $KL(P^{\mathcal{G}}\|GCD)$ and $KL(P^{\mathcal{G}}\|\text{MCMC-T}(k))$ for $k = 10$, for $T \in \{$Uniform, Priority, Restart$\}$. Each point represents a single task. Points below the diagonal indicate tasks where MCMC approximates $P^{\mathcal{G}}$ better than GCD. Fig. 5 displays the same information, but for ASAp$(10)$ instead of GCD.

Fig. 6, Fig. 7, Fig. 8 compare the distance to $P^{\mathcal{G}}$ of the different sampling strategies as compute increases, across all the benchmark tasks. Lower KL-Divergence indicates a better approximation to $P^{\mathcal{G}}$.

The distributions induced by different sampling methods are all approximated using 100 samples, and $P^{\mathcal{G}}$ is approximated using all the samples acquired across all runs for the same task. A $95\%$ confidence band is shown for convergence plots, computed via Bootstrapping.

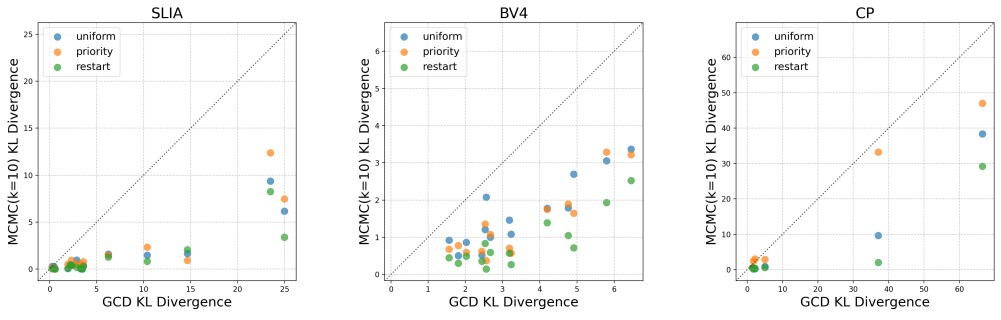

Figure 4: KL-Divergence for GCD vs MCMC($k = 10$) by subset

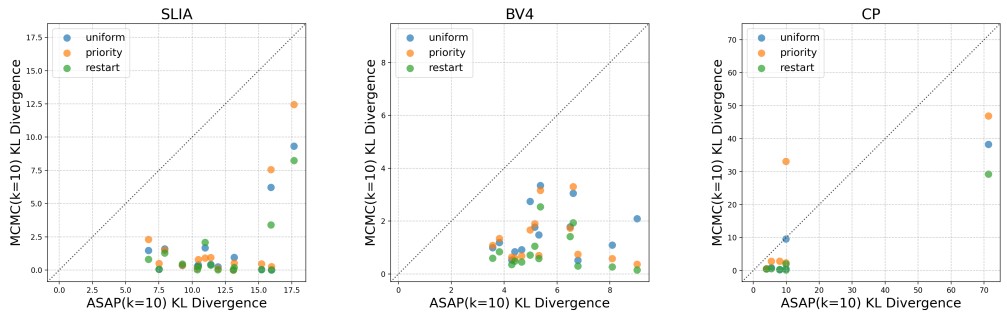

Figure 5: KL-Divergence for ASAp($k = 10$) vs MCMC($k = 10$) by subset

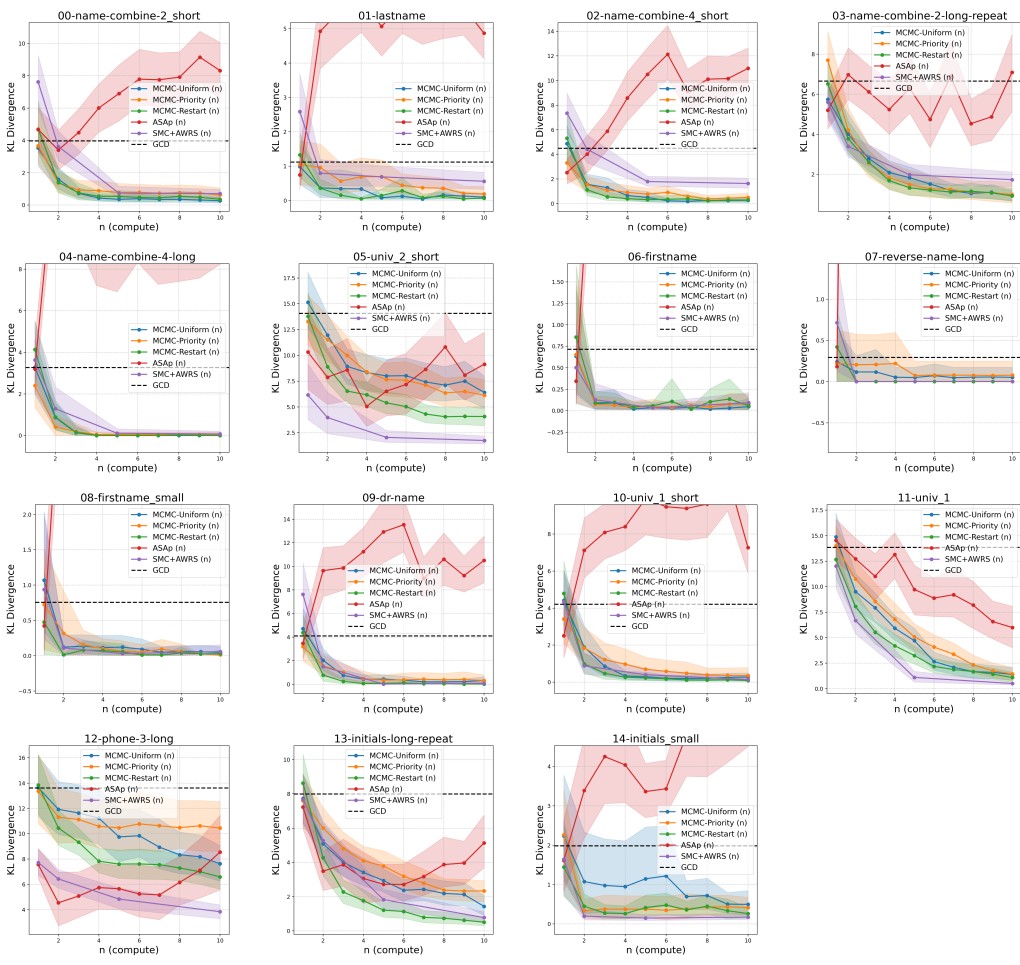

Figure 6: KL-Divergence of sampling methods in SLIA subset

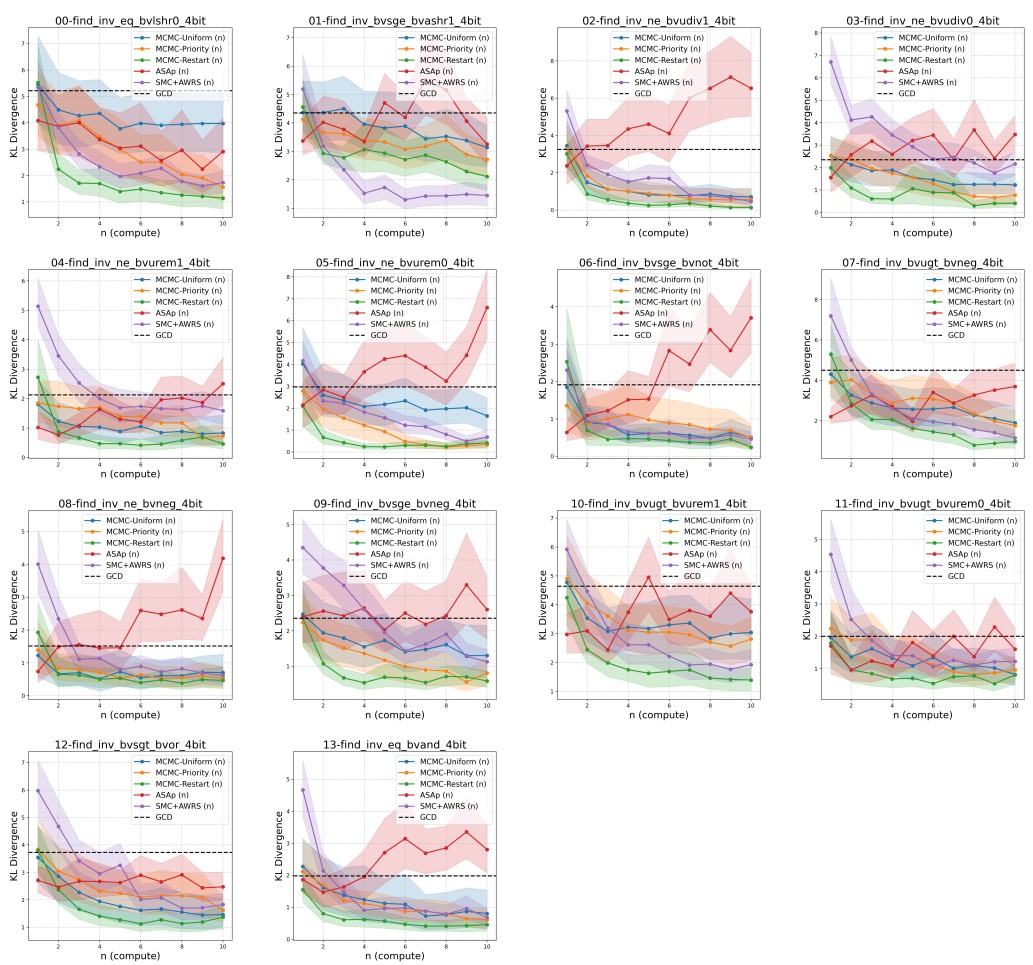

Figure 7: KL-Divergence of sampling methods in BV4 subset

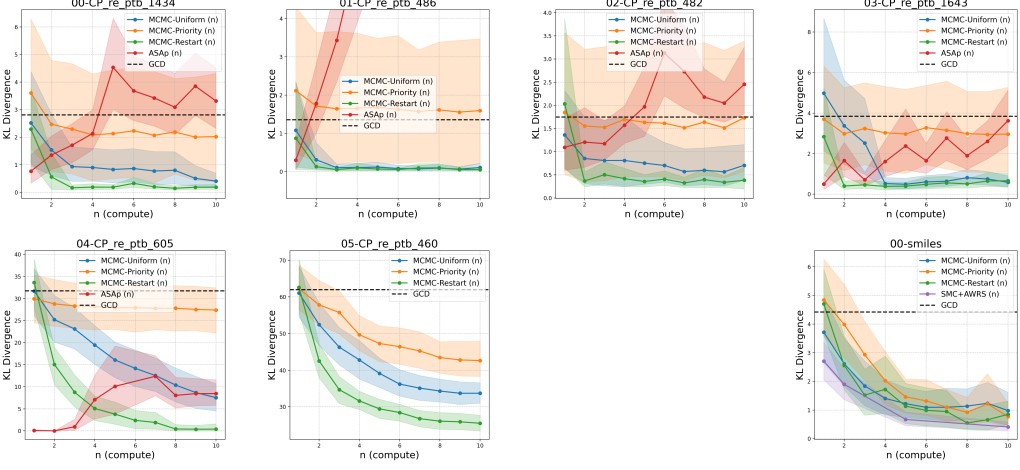

Figure 8: KL-Divergence of sampling methods in CP and MS subset

# E  Fuzzing Experiments Details

## E.1  Benchmarks

Table 1 summarizes the libraries, versions, and seed formats for each target.

Table 1: Benchmarks, versions, and seed formats.

| Target | Library | Version | Seed format |
|--------|---------|---------|-------------|
| XML[22, 21] | libxml2 | 2.15.0 | `.xml` |
| SQL[62] | sqlite | 3.49.2 | `.test` |

## E.2  Parse-Tree Illustration

Figure 9 shows a comprehensive parse tree for a SQLite test case, derived from the grammar in Figure 1b.

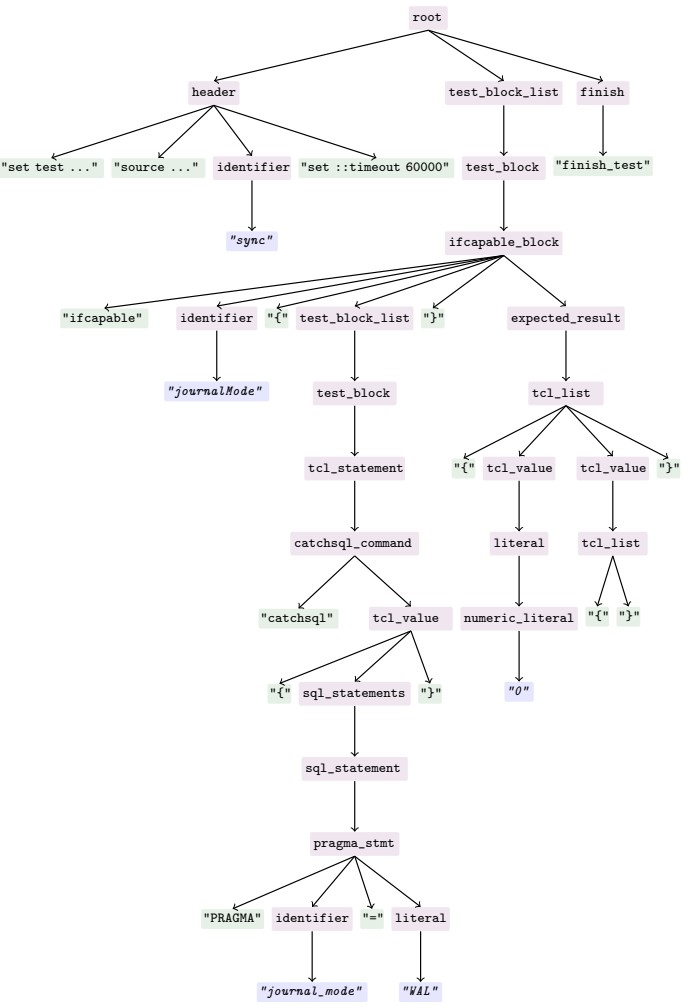

Figure 9: Condensed parse tree for the example SQLite `.test` script used in Figure 1. Purple boxes denote non-terminals, green boxes denote grammar terminals, and blue italics show literal terminal values substituted during this derivation. Subtrees unrelated to the header and the first `do_execsql_test` block are elided for brevity.

## E.3 Prompts and Constraints

For all benchmarks, we use a standard in-context learning format where the prompt consists of a single (specification, solution) pair, followed by a new specification for which the model must generate a solution. A representative prompt for the XML benchmark is shown in Figure 10a, along with its corresponding grammar in Figure 10b.

```
Question 1:
Generate a single, short but complex XML document.
Include a variety of XML features and various
markup declarations. Do not reuse previous solutions.

Solution 1:
<?xml version="1.0" encoding="UTF-8"?>
<!DOCTYPE document [
  <!ENTITY note "Take care!">
]>
<document xmlns="http://example.com/schema"... >
...
</document>

Question 2:
Generate a single, short but complex XML document.
Include a variety of XML features and various
markup declarations. Do not reuse previous solutions.

Solution 2:
```

```
root ::=
    (prolog)? ws?
    (doctype ws)?
    element

prolog ::=
    xmldecl (ws misc)*

...
element ::=
    starttag content endtag
    | emptyelement

endtag ::=
    "</" name ">"

...
content ::=
    (chardata
        | element
        | comment
        | pi
        | cdata)*

comment ::=
    "<!--" [^-]* "-->" ws
...
ws      ::= [white_space]*
```

(a) Prompt                    (b) Grammar

Figure 10: (a) Prompt given to a LM to generate seed test cases for fuzzing the XML parser. (b) Simplified version of the XML grammar written in EBNF notation. The goal of the problem is to generate multiple diverse seeds that trigger different code paths in the library being tested.

## E.4 Fuzzing Protocol and Environment

All fuzzing experiments were conducted using AFL++ 4.00c on the hardware and software setup described in Appendix A. Each (benchmark, method) pair was evaluated in $N = 5$ independent, single-instance AFL++ runs of exactly $21600\,\mathrm{s}$ (six hours). We set 'AFL_RANDOM_SEED' to $42 + i, (i = 1...5)$ for reproducibility and configure standard environment variables to ensure non-interactive execution. All other AFL++ parameters remained at defaults to isolate the impact of seed corpus quality. Complete build and execution scripts are provided in the supplementary materials.

## E.5 Coverage Measurement via LLVM Instrumentation

We measured branch coverage using LLVM's instrumentation toolchain (-fprofile-instr-generate -fcoverage-mapping), which adds $\leq 2\%$ runtime overhead. Raw profiles were collected during execution and aggregated post-trial using llvm-profdata and llvm-cov.

**Rationale.** We report branch coverage rather than crash counts because the experiment isolates *seed quality* — all methods receive the same fixed prompt per benchmark, so coverage is a good measure on how their seeds exercise the code.

### E.6 Rejection Sampling Acceptance Rates

We quantified the viability of rejection sampling under the same grammar constraints used by our MCMC framework in Section 4.2. Across 500 attempted samples per benchmark, the proportion of syntactically and semantically valid outputs was consistently below 1%.

### E.7 Generation Time Analysis

Table 2 reports the average wall-clock time required to generate a single seed across different methods using Llama-3.1-8B-Instruct and Qwen2.5-Coder-7B-Instruct on $1\times$ NVIDIA RTX A6000 GPU.[2] As expected, MCMC with $k = 10$ steps takes approximately $10\times$ longer than GCD per sample, while ASAp is significantly more expensive. This timing difference motivates our compute-matched seed count experiments (Appendix E.8), where we evaluate whether generating more GCD seeds within the same time budget can compensate for MCMC's per-seed quality advantage.

Table 2: Average generation time per seed (seconds) for each method.

(a) Llama-3.1-8B-Instruct

| Method | $k$ | XML | SQL |
|---|---|---|---|
| **MCMC-Uniform** | | | |
| | 2 | 14 | 21 |
| | 5 | 36 | 56 |
| | 10 | 71 | 112 |
| **MCMC-Priority** | | | |
| | 2 | 14 | 21 |
| | 5 | 36 | 56 |
| | 10 | 71 | 112 |
| **MCMC-Restart** | | | |
| | 2 | 14 | 21 |
| | 5 | 36 | 56 |
| | 10 | 71 | 112 |
| **SMC+AWRS** | | | |
| | 2 | 31 | 46 |
| | 5 | 111 | 158 |
| | 10 | 161 | 214 |
| **ASAp** | 10 | 461 | 714 |
| **GCD** | — | 7 | 11 |

(b) Qwen2.5-Coder-7B-Instruct

| Method | $k$ | XML | SQL |
|---|---|---|---|
| **MCMC-Uniform** | | | |
| | 2 | 12 | 18 |
| | 5 | 31 | 48 |
| | 10 | 63 | 98 |
| **MCMC-Priority** | | | |
| | 2 | 12 | 18 |
| | 5 | 31 | 49 |
| | 10 | 64 | 99 |
| **MCMC-Restart** | | | |
| | 2 | 12 | 18 |
| | 5 | 32 | 49 |
| | 10 | 64 | 100 |
| **SMC+AWRS** | | | |
| | 2 | 27 | 40 |
| | 5 | 96 | 138 |
| | 10 | 142 | 188 |
| **ASAp** | 10 | 402 | 628 |
| **GCD** | — | 6 | 9 |

### E.8 Seed Count Ablation

To evaluate whether seed quality or quantity matters more for fuzzing effectiveness, we conducted compute-matched experiments varying the number of initial seeds $N \in \{50, 100, 200, 500\}$. Since MCMC (10) takes approximately $10\times$ longer than GCD per seed (Appendix E.7), comparing 50 MCMC seeds against 500 GCD seeds represents equivalent computational budgets.

Tables 3 and 4 show branch coverage after 6-hour fuzzing runs for both Llama-3.1-8B-Instruct and Qwen2.5-Coder-7B-Instruct. Remarkably, MCMC methods with just 50 seeds consistently outperform GCD with 500 seeds, demonstrating that sampling fidelity dominates sheer quantity for fuzzing effectiveness.

### E.9 Branch Coverage and KL Divergence Across Models

Figures 11 and 12 show branch coverage and KL divergence results for both Llama-3.1-8B-Instruct and Qwen2.5-Coder-7B-Instruct on the SQL and XML benchmarks, respectively. The trends observed in the main text (Figure 3 for SQL with Llama-3.1-8B-Instruct) hold consistently across both models:

---

[2]SMC+AWRS ($k = 10$) was evaluated on $1\times$ H100 GPU (120GB) due to higher memory requirements.

Table 3: SQL benchmark: Branch coverage (mean ± std) over five trials after 6-hour fuzzing runs with varying seed counts. Best entries per seed count in green.

| Method | $k$ | N=50 | N=100 | N=200 | N=500 |
|---|---|---|---|---|---|
| *Llama-3.1-8B-Instruct* | | | | | |
| **MCMC-Uniform** | | | | | |
| | 2 | $30.72 \pm 0.60$ | $33.69 \pm 0.66$ | $33.41 \pm 0.66$ | $33.39 \pm 0.66$ |
| | 5 | $32.07 \pm 0.45$ | $34.37 \pm 0.48$ | $34.25 \pm 0.48$ | $35.53 \pm 0.49$ |
| | 10 | $33.88 \pm 0.64$ | $36.47 \pm 0.69$ | $35.49 \pm 0.67$ | $37.03 \pm 0.70$ |
| **MCMC-Priority** | | | | | |
| | 2 | $32.04 \pm 1.16$ | $33.89 \pm 1.23$ | $33.95 \pm 1.23$ | $33.98 \pm 1.23$ |
| | 5 | $33.10 \pm 0.25$ | $36.21 \pm 0.27$ | $36.18 \pm 0.27$ | $36.01 \pm 0.27$ |
| | 10 | $34.48 \pm 0.74$ | $36.69 \pm 0.78$ | $37.02 \pm 0.79$ | $37.17 \pm 0.79$ |
| **MCMC-Restart** | | | | | |
| | 2 | $31.12 \pm 0.46$ | $33.38 \pm 0.49$ | $33.09 \pm 0.49$ | $33.56 \pm 0.50$ |
| | 5 | $32.22 \pm 0.86$ | $33.36 \pm 0.90$ | $34.63 \pm 0.93$ | $34.08 \pm 0.91$ |
| | 10 | $33.05 \pm 0.68$ | $36.25 \pm 0.74$ | $36.81 \pm 0.75$ | $36.64 \pm 0.75$ |
| **SMC+AWRS** | | | | | |
| | 2 | $33.20 \pm 1.09$ | $32.93 \pm 1.08$ | $33.71 \pm 1.11$ | $33.39 \pm 1.10$ |
| | 5 | $34.63 \pm 0.43$ | $35.16 \pm 0.44$ | $34.51 \pm 0.43$ | $35.74 \pm 0.45$ |
| | 10 | $35.85 \pm 0.75$ | $36.44 \pm 0.77$ | $35.93 \pm 0.76$ | $36.23 \pm 0.76$ |
| **ASAp** | 10 | $33.09 \pm 0.55$ | $32.61 \pm 0.54$ | $33.65 \pm 0.56$ | $33.39 \pm 0.55$ |
| **GCD** | — | $30.47 \pm 0.91$ | $31.52 \pm 0.94$ | $31.37 \pm 0.94$ | $31.70 \pm 0.95$ |
| **Grammarinator** | — | $25.47 \pm 0.49$ | $26.50 \pm 0.51$ | $28.39 \pm 0.55$ | $27.92 \pm 0.54$ |
| *Qwen2.5-Coder-7B-Instruct* | | | | | |
| **MCMC-Uniform** | | | | | |
| | 2 | $30.85 \pm 0.61$ | $32.79 \pm 0.64$ | $33.65 \pm 0.66$ | $33.07 \pm 0.65$ |
| | 5 | $32.35 \pm 0.45$ | $34.91 \pm 0.49$ | $34.48 \pm 0.48$ | $35.45 \pm 0.49$ |
| | 10 | $33.29 \pm 0.63$ | $36.87 \pm 0.70$ | $36.70 \pm 0.69$ | $35.91 \pm 0.68$ |
| **MCMC-Priority** | | | | | |
| | 2 | $31.87 \pm 1.15$ | $34.72 \pm 1.26$ | $34.80 \pm 1.26$ | $34.87 \pm 1.26$ |
| | 5 | $34.67 \pm 0.26$ | $35.56 \pm 0.27$ | $36.21 \pm 0.27$ | $35.89 \pm 0.27$ |
| | 10 | $35.80 \pm 0.77$ | $36.41 \pm 0.80$ | $37.22 \pm 0.80$ | $36.85 \pm 0.79$ |
| **MCMC-Restart** | | | | | |
| | 2 | $31.30 \pm 0.46$ | $32.77 \pm 0.48$ | $33.31 \pm 0.49$ | $33.36 \pm 0.49$ |
| | 5 | $32.98 \pm 0.89$ | $33.94 \pm 0.91$ | $33.46 \pm 0.90$ | $34.45 \pm 0.92$ |
| | 10 | $34.38 \pm 0.70$ | $35.69 \pm 0.73$ | $35.99 \pm 0.74$ | $35.79 \pm 0.73$ |
| **SMC+AWRS** | | | | | |
| | 2 | $33.50 \pm 1.10$ | $33.37 \pm 1.10$ | $32.83 \pm 1.08$ | $32.94 \pm 1.08$ |
| | 5 | $34.50 \pm 0.43$ | $35.35 \pm 0.44$ | $34.90 \pm 0.44$ | $35.17 \pm 0.44$ |
| | 10 | $36.79 \pm 0.77$ | $35.84 \pm 0.75$ | $36.07 \pm 0.76$ | $36.57 \pm 0.77$ |
| **ASAp** | 10 | $32.67 \pm 0.54$ | $32.47 \pm 0.54$ | $32.75 \pm 0.54$ | $32.58 \pm 0.54$ |
| **GCD** | — | $29.51 \pm 0.88$ | $31.41 \pm 0.94$ | $31.44 \pm 0.94$ | $31.49 \pm 0.94$ |
| **Grammarinator** | — | $25.47 \pm 0.49$ | $26.50 \pm 0.51$ | $28.39 \pm 0.55$ | $27.92 \pm 0.54$ |

(i) KL divergence decreases with more MCMC steps, indicating convergence toward the target distribution, and (ii) MCMC-based methods achieve higher branch coverage than GCD and ASAp, with MCMC-Priority ($k = 10$) delivering the best performance. Qwen2.5-Coder-7B-Instruct exhibits similar relative performance across methods, though with slightly different coverage values due to model-specific generation characteristics.

## E.10 Overall Coverage Results

Tables 5 and 6 show that the improvements achieved by our grammar-based MCMC sampler on branch coverage translate consistently to both function and line coverage across both the SQL and XML benchmarks using 100 seeds over 6-hour fuzzing runs. Coverage grows monotonically with

Table 4: XML benchmark: Branch coverage (mean ± std) over five trials after 6-hour fuzzing runs with varying seed counts. Best entries per seed count in green.

| Method | $k$ | N=50 | N=100 | N=200 | N=500 |
|---|---|---|---|---|---|
| *Llama-3.1-8B-Instruct* | | | | | |
| **MCMC-Uniform** | | | | | |
| | 2 | $12.41 \pm 0.15$ | $12.51 \pm 0.16$ | $12.45 \pm 0.16$ | $12.43 \pm 0.15$ |
| | 5 | $12.29 \pm 0.13$ | $12.45 \pm 0.13$ | $12.88 \pm 0.14$ | $12.99 \pm 0.14$ |
| | 10 | $12.47 \pm 0.04$ | $12.76 \pm 0.05$ | $12.97 \pm 0.05$ | $12.96 \pm 0.05$ |
| **MCMC-Priority** | | | | | |
| | 2 | $11.91 \pm 0.06$ | $12.60 \pm 0.07$ | $12.96 \pm 0.07$ | $12.68 \pm 0.07$ |
| | 5 | $12.13 \pm 0.07$ | $12.82 \pm 0.07$ | $12.70 \pm 0.07$ | $13.05 \pm 0.07$ |
| | 10 | $12.52 \pm 0.27$ | $12.79 \pm 0.28$ | $12.94 \pm 0.28$ | $12.88 \pm 0.28$ |
| **MCMC-Restart** | | | | | |
| | 2 | $11.51 \pm 0.08$ | $11.70 \pm 0.08$ | $11.99 \pm 0.08$ | $12.07 \pm 0.09$ |
| | 5 | $11.69 \pm 0.05$ | $12.11 \pm 0.05$ | $12.44 \pm 0.06$ | $12.24 \pm 0.06$ |
| | 10 | $12.08 \pm 0.07$ | $12.58 \pm 0.08$ | $12.85 \pm 0.08$ | $12.61 \pm 0.08$ |
| **SMC+AWRS** | | | | | |
| | 2 | $12.60 \pm 0.08$ | $12.54 \pm 0.08$ | $12.38 \pm 0.08$ | $12.63 \pm 0.08$ |
| | 5 | $12.60 \pm 0.04$ | $12.53 \pm 0.04$ | $12.73 \pm 0.04$ | $12.68 \pm 0.04$ |
| | 10 | $12.54 \pm 0.13$ | $12.63 \pm 0.13$ | $12.86 \pm 0.13$ | $12.79 \pm 0.13$ |
| **ASAp** | 10 | $11.06 \pm 0.08$ | $11.30 \pm 0.08$ | $11.34 \pm 0.08$ | $11.04 \pm 0.08$ |
| **GCD** | — | $11.19 \pm 0.07$ | $11.08 \pm 0.07$ | $11.36 \pm 0.07$ | $11.27 \pm 0.07$ |
| **Grammarinator** | — | $9.01 \pm 0.16$ | $8.99 \pm 0.16$ | $9.21 \pm 0.16$ | $8.94 \pm 0.16$ |
| *Qwen2.5-Coder-7B-Instruct* | | | | | |
| **MCMC-Uniform** | | | | | |
| | 2 | $12.20 \pm 0.15$ | $12.13 \pm 0.15$ | $12.69 \pm 0.16$ | $12.76 \pm 0.16$ |
| | 5 | $12.33 \pm 0.13$ | $12.57 \pm 0.13$ | $12.82 \pm 0.14$ | $12.78 \pm 0.13$ |
| | 10 | $12.76 \pm 0.05$ | $12.65 \pm 0.05$ | $12.74 \pm 0.05$ | $13.26 \pm 0.05$ |
| **MCMC-Priority** | | | | | |
| | 2 | $12.81 \pm 0.07$ | $12.75 \pm 0.07$ | $12.73 \pm 0.07$ | $12.77 \pm 0.07$ |
| | 5 | $12.76 \pm 0.07$ | $12.92 \pm 0.07$ | $13.04 \pm 0.07$ | $13.03 \pm 0.07$ |
| | 10 | $12.89 \pm 0.28$ | $12.68 \pm 0.27$ | $12.84 \pm 0.28$ | $13.27 \pm 0.29$ |
| **MCMC-Restart** | | | | | |
| | 2 | $11.93 \pm 0.08$ | $11.69 \pm 0.08$ | $11.83 \pm 0.08$ | $12.26 \pm 0.09$ |
| | 5 | $11.94 \pm 0.05$ | $12.05 \pm 0.05$ | $12.35 \pm 0.06$ | $12.58 \pm 0.06$ |
| | 10 | $12.62 \pm 0.08$ | $12.45 \pm 0.08$ | $12.80 \pm 0.08$ | $12.73 \pm 0.08$ |
| **SMC+AWRS** | | | | | |
| | 2 | $12.43 \pm 0.08$ | $12.64 \pm 0.08$ | $12.59 \pm 0.08$ | $12.69 \pm 0.08$ |
| | 5 | $12.74 \pm 0.04$ | $12.70 \pm 0.04$ | $12.46 \pm 0.04$ | $12.60 \pm 0.04$ |
| | 10 | $12.83 \pm 0.13$ | $12.82 \pm 0.13$ | $13.01 \pm 0.13$ | $12.90 \pm 0.13$ |
| **ASAp** | 10 | $11.44 \pm 0.08$ | $11.32 \pm 0.08$ | $11.39 \pm 0.08$ | $11.26 \pm 0.08$ |
| **GCD** | — | $11.19 \pm 0.07$ | $11.39 \pm 0.07$ | $11.58 \pm 0.07$ | $11.99 \pm 0.07$ |
| **Grammarinator** | — | $9.01 \pm 0.16$ | $8.99 \pm 0.16$ | $9.21 \pm 0.16$ | $8.94 \pm 0.16$ |

the number of steps $k$; however, $k = 5$ already captures $\geq 95\%$ of the gain realized by $k = 10$ on both benchmarks. Even the weakest setting ($k = 2$) surpasses GCD's final coverage by 4-6%, demonstrating that MCMC proposals yield coverage gains over heuristic constrained decoding with very few sampling steps. Additionally, seed count ablation results (Appendix E.8) and generation timing analysis (Appendix E.7) demonstrate that MCMC's quality advantages persist even under compute-matched comparisons.

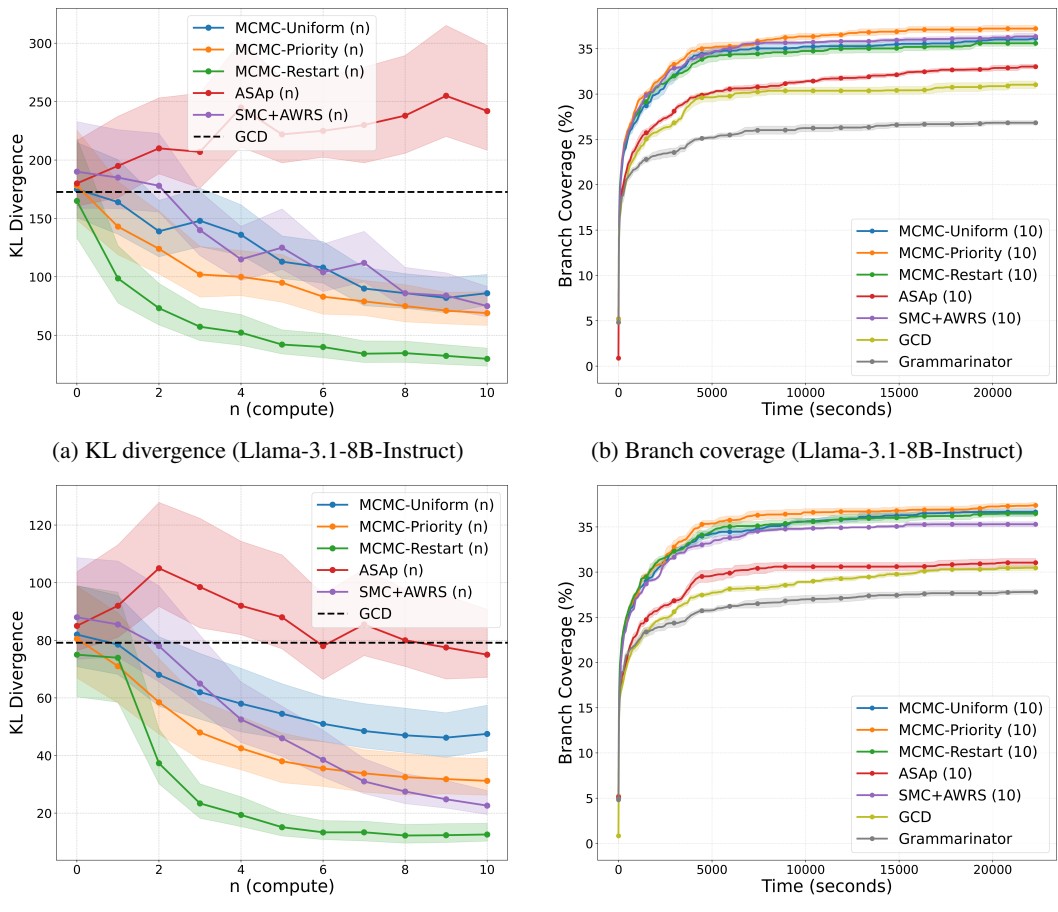

(a) KL divergence (Llama-3.1-8B-Instruct)

(b) Branch coverage (Llama-3.1-8B-Instruct)

(c) KL divergence (Qwen2.5-Coder-7B-Instruct)

(d) Branch coverage (Qwen2.5-Coder-7B-Instruct)

Figure 11: SQL benchmark: KL divergence and branch coverage over time for both language models (100 seeds, 6-hour runs). Top row shows Llama-3.1-8B-Instruct results; bottom row shows Qwen2.5-Coder-7B-Instruct results.

# F  Properties and Proofs

In this section, we formalize and prove the two key properties of our sampler (Alg. 1), constraint satisfying (Thm. 1) and monotonically converging (Thm. 3).

The first property follows directly from the procedure in Alg. 1.

**Theorem 1** (Constraint Satisfying). *For any LM $P$, any grammar $G$, any chain length $k$, and any truncation distribution $p_{\text{POS}}$, the result of Alg. 1 is always inside $\mathcal{L}(G)$.*

*Proof.* The output of Alg. 1 can only be generated from either Line 1 or Line 11, both of which call the GCD procedure. Since GCD samples only from the constrained language $\mathcal{L}(G)$, the result of Alg. 1 must also fall within $\mathcal{L}(G)$. $\qquad\square$

As for monotonically converging, we prove it by applying the following theorem for Markov chains.

**Theorem 2** (Thm. 5.6.6 in [15]). *Let $p$ be a Markov chain with countable states, and let $\|q, q'\|_{\text{TV}}$ denotes the total variance distance of two distributions, i.e., $\frac{1}{2}\sum_x |q(x) - q'(x)|$.*

*When $p$ is irreducible, aperiodic, and has stationary distribution $\pi$, then for any state state $x$ $\|p^k(\cdot \mid x), \pi\|_{\text{TV}}$ will converge to $0$ as $k$ approaches to $\infty$.*

**Theorem 3** (Monotonically Converging). *For any LM $P$, any grammar $G$, and any truncation distribution $p_{\text{POS}}$, if $p_{\text{POS}}^w(0) > 0$ for all sequences $w \in \mathcal{L}(G)$, then the output distribution of Alg. 1*

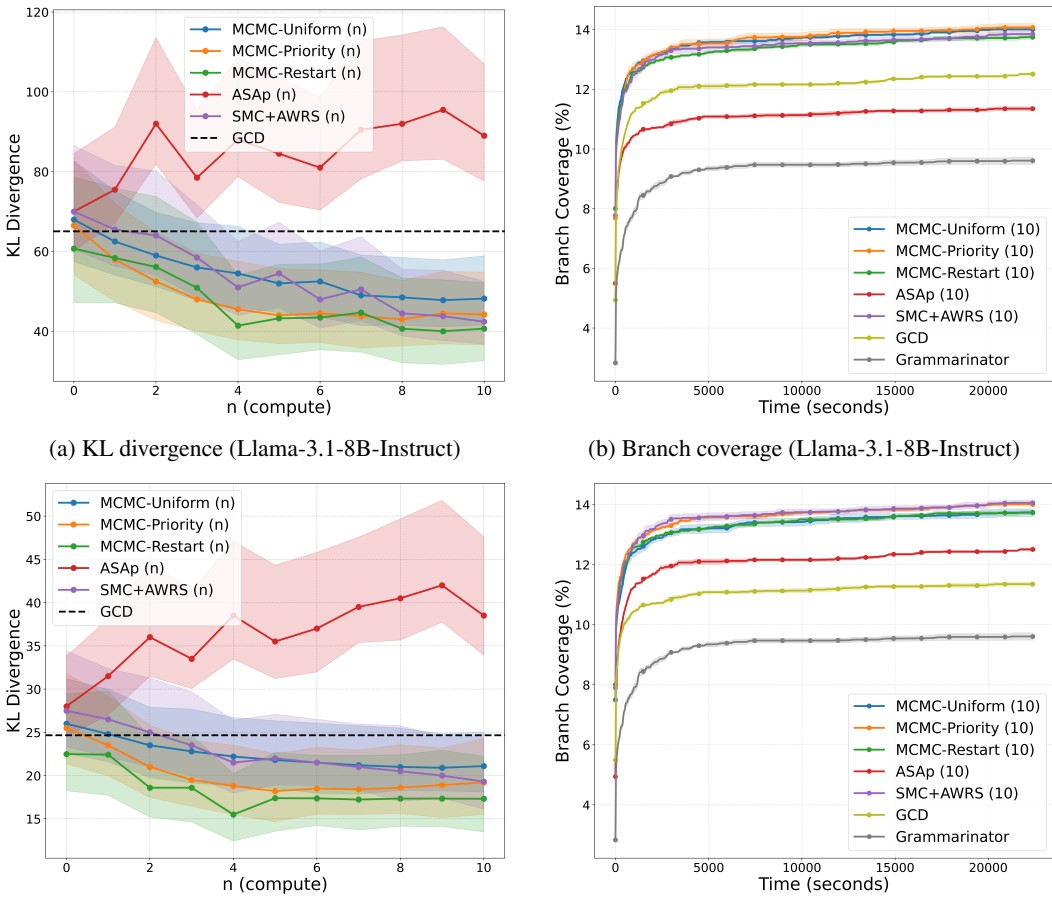

(a) KL divergence (Llama-3.1-8B-Instruct)

(b) Branch coverage (Llama-3.1-8B-Instruct)

(c) KL divergence (Qwen2.5-Coder-7B-Instruct)

(d) Branch coverage (Qwen2.5-Coder-7B-Instruct)

Figure 12: XML benchmark: KL divergence and branch coverage over time for both language models (100 seeds, 6-hour runs). Top row shows Llama-3.1-8B-Instruct results; bottom row shows Qwen2.5-Coder-7B-Instruct results.

*will monotonically converge to $P^{\mathcal{G}}$ as the chain length $k$ approaches to infinite, as shown below.*

$$\lim_{k \to \infty} \|P_k^{\mathcal{O}}, P^{\mathcal{G}}\|_{\text{TV}} = 0 \tag{3}$$

$$\forall k, \|P_k^{\mathcal{O}}, P^{\mathcal{G}}\|_{\text{TV}} \geq \|P_{k+1}^{\mathcal{O}}, P^{\mathcal{G}}\|_{\text{TV}} \tag{4}$$

*where $P_k^{\mathcal{O}}$ denotes the output distribution of Alg. 1 when the chain length is $k$.*

*Proof.* We prove the **convergence** (Eq. 3) by applying Thm. 2 to our case, by verifying that the Markov chain constructed in Alg. 1 satisfies all prerequisites of Thm. 2.

1. (Countable states) In our Markov chain, the state set comprises all sequences with non-zero probability in $P^{\mathcal{G}}$, denoted as $S$. This set is countable since the set of all sequences is countable.

2. (Irreducibility) Let $q$ be the proposal distribution of our Markov chain. We start by showing that $q(y \mid x) > 0$ for all $x, y \in S$. Consider the event where the empty prefix is selected in Line 10, and $y$ is selected by GCD in Line 11. The probability of this event is $p_{\text{POS}}^x(0) \cdot P_{\text{GCD}}(y)$, which is non-zero. On the other hand, $q(y \mid x)$ is no smaller than this probability, hence it must also be non-zero.

Table 5: SQL benchmark coverage (mean ± std) over five trials after 6-hour fuzzing runs with 100 seeds. Best entries highlighted in green.

| Method | $k$ | Branch (%) | Function (%) | Line (%) |
|---|---|---|---|---|
| *Llama-3.1-8B-Instruct* | | | | |
| **MCMC-Uniform** | | | | |
| | 2 | 33.69 ± 0.66 | 52.17 ± 0.53 | 42.07 ± 0.32 |
| | 5 | 34.37 ± 0.48 | 56.25 ± 0.76 | 44.51 ± 0.37 |
| | 10 | 36.47 ± 0.69 | 57.44 ± 0.89 | 45.98 ± 0.79 |
| **MCMC-Priority** | | | | |
| | 2 | 33.89 ± 1.29 | 53.48 ± 1.60 | 43.32 ± 1.59 |
| | 5 | 36.21 ± 0.27 | 57.27 ± 0.86 | 45.96 ± 0.45 |
| | 10 | 36.69 ± 0.78 | 58.87 ± 0.82 | 47.18 ± 1.02 |
| **MCMC-Restart** | | | | |
| | 2 | 33.38 ± 0.49 | 52.50 ± 0.78 | 42.17 ± 0.74 |
| | 5 | 33.36 ± 0.90 | 55.03 ± 0.99 | 43.50 ± 0.92 |
| | 10 | 36.25 ± 0.74 | 56.31 ± 0.76 | 45.24 ± 0.71 |
| **SMC+AWRS** | | | | |
| | 2 | 32.93 ± 1.08 | 53.01 ± 1.26 | 43.05 ± 0.96 |
| | 5 | 35.16 ± 0.44 | 57.10 ± 0.65 | 45.79 ± 0.44 |
| | 10 | 36.44 ± 0.77 | 58.70 ± 0.57 | 46.76 ± 0.54 |
| **ASAp** | 10 | 32.61 ± 0.54 | 52.69 ± 0.94 | 43.26 ± 0.74 |
| **GCD** | — | 31.52 ± 0.94 | 49.38 ± 0.60 | 39.25 ± 0.65 |
| **Grammarinator** | — | 26.50 ± 0.51 | 44.80 ± 1.09 | 35.66 ± 1.20 |
| *Qwen2.5-Coder-7B-Instruct* | | | | |
| **MCMC-Uniform** | | | | |
| | 2 | 32.79 ± 0.64 | 49.62 ± 0.56 | 39.93 ± 0.43 |
| | 5 | 34.91 ± 0.49 | 56.07 ± 0.72 | 44.21 ± 0.29 |
| | 10 | 36.87 ± 0.70 | 57.61 ± 0.71 | 46.08 ± 1.14 |
| **MCMC-Priority** | | | | |
| | 2 | 34.72 ± 1.26 | 51.66 ± 1.30 | 41.90 ± 1.69 |
| | 5 | 35.56 ± 0.27 | 55.67 ± 0.73 | 44.87 ± 0.38 |
| | 10 | 36.41 ± 0.80 | 58.27 ± 0.62 | 47.41 ± 0.86 |
| **MCMC-Restart** | | | | |
| | 2 | 32.77 ± 0.48 | 50.98 ± 0.77 | 41.02 ± 0.64 |
| | 5 | 33.94 ± 0.91 | 54.37 ± 1.03 | 42.99 ± 0.89 |
| | 10 | 35.69 ± 0.73 | 57.46 ± 0.96 | 46.29 ± 0.82 |
| **SMC+AWRS** | | | | |
| | 2 | 33.37 ± 1.10 | 49.75 ± 1.00 | 40.27 ± 0.92 |
| | 5 | 35.35 ± 0.44 | 55.70 ± 0.50 | 44.67 ± 0.48 |
| | 10 | 35.84 ± 0.75 | 57.00 ± 0.52 | 45.56 ± 0.65 |
| **ASAp** | 10 | 32.47 ± 0.54 | 49.05 ± 0.74 | 39.17 ± 0.77 |
| **GCD** | — | 31.41 ± 0.94 | 48.32 ± 1.02 | 39.41 ± 0.55 |
| **Grammarinator** | — | 26.50 ± 0.51 | 44.80 ± 1.09 | 35.66 ± 1.20 |

Then, by the definition of the transition probability $p$ in the Metropolis-Hastings algorithm[3]

$$\forall x, y \in S, \quad p(y \mid x) \geq q(y \mid x) \cdot \alpha(x, y) = q(y \mid x) \cdot \max \left\{ 1, \frac{P(y)q(x \mid y)}{P(x)q(y \mid x)} \right\}.$$

All values on the right-hand side are positive, so $p(y \mid x)$ must also be positive, implying the irreducibility of the Markov chain.

3. (Aperiodicity) By the above analysis, $p(x \mid x) > 0$ for any state $x$, implying aperiodicity.

---

[3]The greater-than part of the first inequality captures the special case where $x = y$.

Table 6: XML benchmark coverage (mean $\pm$ std) over five trials after 6-hour fuzzing runs with 100 seeds. Best entries highlighted in green.

| Method | $k$ | Branch (%) | Function (%) | Line (%) |
|---|---|---|---|---|
| | | *Llama-3.1-8B-Instruct* | | |
| **MCMC-Uniform** | | | | |
| | 2 | $12.51 \pm 0.16$ | $18.89 \pm 0.16$ | $13.59 \pm 0.09$ |
| | 5 | $12.45 \pm 0.13$ | $19.23 \pm 0.06$ | $13.82 \pm 0.10$ |
| | 10 | $12.76 \pm 0.05$ | $19.28 \pm 0.05$ | $13.98 \pm 0.03$ |
| **MCMC-Priority** | | | | |
| | 2 | $12.60 \pm 0.07$ | $19.21 \pm 0.05$ | $14.05 \pm 0.06$ |
| | 5 | $12.82 \pm 0.07$ | $19.26 \pm 0.08$ | $13.99 \pm 0.10$ |
| | 10 | $12.79 \pm 0.28$ | $19.44 \pm 0.37$ | $14.22 \pm 0.28$ |
| **MCMC-Restart** | | | | |
| | 2 | $11.70 \pm 0.08$ | $18.46 \pm 0.22$ | $13.23 \pm 0.10$ |
| | 5 | $12.11 \pm 0.05$ | $18.62 \pm 0.04$ | $13.45 \pm 0.05$ |
| | 10 | $12.58 \pm 0.08$ | $19.26 \pm 0.19$ | $13.79 \pm 0.09$ |
| **SMC+AWRS** | | | | |
| | 2 | $12.54 \pm 0.08$ | $19.18 \pm 0.08$ | $13.85 \pm 0.04$ |
| | 5 | $12.53 \pm 0.04$ | $19.36 \pm 0.06$ | $13.96 \pm 0.06$ |
| | 10 | $12.63 \pm 0.13$ | $19.42 \pm 0.18$ | $14.21 \pm 0.11$ |
| **ASAp** | 10 | $11.30 \pm 0.08$ | $17.54 \pm 0.02$ | $12.53 \pm 0.04$ |
| **GCD** | — | $11.08 \pm 0.07$ | $17.21 \pm 0.03$ | $12.41 \pm 0.07$ |
| **Grammarinator** | — | $8.99 \pm 0.16$ | $15.18 \pm 0.29$ | $10.60 \pm 0.14$ |
| | | *Qwen2.5-Coder-7B-Instruct* | | |
| **MCMC-Uniform** | | | | |
| | 2 | $12.13 \pm 0.15$ | $20.77 \pm 0.23$ | $15.13 \pm 0.18$ |
| | 5 | $12.57 \pm 0.13$ | $21.23 \pm 0.13$ | $15.45 \pm 0.15$ |
| | 10 | $12.65 \pm 0.05$ | $20.53 \pm 0.42$ | $15.13 \pm 0.22$ |
| **MCMC-Priority** | | | | |
| | 2 | $12.75 \pm 0.07$ | $21.63 \pm 0.06$ | $15.80 \pm 0.06$ |
| | 5 | $12.92 \pm 0.07$ | $21.24 \pm 0.12$ | $15.60 \pm 0.16$ |
| | 10 | $12.68 \pm 0.27$ | $20.81 \pm 0.11$ | $15.51 \pm 0.09$ |
| **MCMC-Restart** | | | | |
| | 2 | $11.69 \pm 0.08$ | $19.86 \pm 0.26$ | $14.32 \pm 0.11$ |
| | 5 | $12.05 \pm 0.05$ | $19.89 \pm 0.08$ | $14.60 \pm 0.13$ |
| | 10 | $12.45 \pm 0.08$ | $20.90 \pm 0.23$ | $14.98 \pm 0.13$ |
| **SMC+AWRS** | | | | |
| | 2 | $12.64 \pm 0.08$ | $21.46 \pm 0.09$ | $15.61 \pm 0.14$ |
| | 5 | $12.70 \pm 0.04$ | $21.23 \pm 0.05$ | $15.32 \pm 0.10$ |
| | 10 | $12.82 \pm 0.13$ | $21.04 \pm 0.24$ | $15.38 \pm 0.09$ |
| **ASAp** | 10 | $11.32 \pm 0.08$ | $18.83 \pm 0.09$ | $13.80 \pm 0.10$ |
| **GCD** | — | $11.39 \pm 0.07$ | $17.46 \pm 0.06$ | $12.54 \pm 0.12$ |
| **Grammarinator** | — | $8.99 \pm 0.16$ | $15.18 \pm 0.29$ | $10.60 \pm 0.14$ |

4. (Stationary distribution) The Metropolis-Hastings algorithm ensures that the target distribution $P^{\mathcal{G}}$ is a stationary distribution of the constructed Markov chain.

Hence, all prerequisites of Thm. 2 are satisfied, then Eq. 3 follows directly from it.

Then, we prove the **monotocity** by the following derivation, where $p(w \mid w')$ denotes the probability for our Markov chain to move from $w'$ to $w$.

$$\|P_{k+1}^{\mathcal{O}}, P^{\mathcal{G}}\|_{\text{TV}} = \frac{1}{2} \sum_w \left| P_{k+1}^{\mathcal{O}}(w) - P^{\mathcal{G}}(w) \right|$$

$$= \frac{1}{2} \sum_w \left| \sum_{w'} \left( P_k^{\mathcal{O}}(w') - P^{\mathcal{G}}(w') \right) \cdot p(w \mid w') \right|$$

$$\leq \frac{1}{2} \sum_w \sum_{w'} \left| P_k^{\mathcal{O}}(w') - P^{\mathcal{G}}(w') \right| \cdot p(w \mid w')$$

$$= \frac{1}{2} \sum_{w'} \left| P_k^{\mathcal{O}}(w') - P^{\mathcal{G}}(w') \right| \left( \sum_w p(w \mid w') \right)$$

$$= \frac{1}{2} \sum_{w'} \left| P_k^{\mathcal{O}}(w') - P^{\mathcal{G}}(w') \right| = \|P_k^{\mathcal{O}}, P^{\mathcal{G}}\|_{\text{TV}}$$

$\square$

