# OpenReview forum: "Constrained Sampling for Language Models Should Be Easy: An MCMC Perspective"
_NeurIPS.cc/2025/Conference — NeurIPS 2025 poster_

### Official Review · Reviewer_4jfZ · 2025-06-23

**Clarity:** 3
**Significance:** 2
**Originality:** 2
**Rating:** 5
**Confidence:** 5

**Summary:**

The authors present a general approach to conditionally sampling from language models subject to constraints (here specifically membership in a context-free language). They note drawbacks with existing approaches: rejection sampling yields valid samples but runtime scales unfavorably when the marginal likelihood of satisfying the constraint under the unconditional model is low, grammar-constrained decoding (GCD) yields constraint-satisfying samples by design, but distorts the distribution over completed strings. They propose a simple MCMC algorithm leveraging GCD in the proposal distribution (with a few different methods attempted for editing sampled traces), then accept/reject edits via MH using the LM likelihood. This yields a correct inference algorithm targeting the true conditional posterior as time grows. The authors present both intrinsic results showing that this procedure empirically begins to converge within a reasonably small number of steps by showing a decrease in a Monte Carlo KL estimator. They also present extrinsic quality results, finding that this approach yields more diverse branch coverage in a downstream program fuzzing task.

**Questions:**

Can you include some plots/tables of how long it takes to generate the seed samples across the different conditions? Since it presumably takes ~10x longer to produce 100 samples from the MCMC (k=10) condition than GCD, using 100 seed samples from each seems to be an odd comparison as MCMC spends considerably more time refining each sample. A more fair comparison might be to produce however many K samples you can from each condition in some fixed time budget then use all those seeds.

**Ethical Concerns:**

["NO or VERY MINOR ethics concerns only"]

**Final Justification:**

This is an important topic area with a clear problem presentation and sound approach. The new analyses presented during reviews cover my key concerns surrounding inadequate baselines and discussion of related work. Should these additions be incorporated, and the paper narrative updated to put the results into appropriate context, I recommend acceptance of this paper.

**Limitations:**

Yes

**Quality:**

3

**Strengths And Weaknesses:**

Strengths:
- The paper is well written and the problem presentation is clear. This is an important issue worth focusing on. The algorithm is a reasonable candidate solution, and the evaluation domain is well-suited for the goal of measuring diversity.
- I like the comparison between the uniform, priority, and restart proposals.

Weaknesses:
- The paper neglects a body of highly related work, missing high-performing baselines. Loula et al (2024), Ahmed et al (2024), and Lipkin et al (2025) also presented this same problem of GCD distortion. All these works presented solutions using GCD as proposal distributions that are then updated using sequential importance sampling and resampling. The methods presented in these papers yield a different set of asymptotic guarantees (convergence in number of particles rather than time), yet can be parallelized as opposed to sequential editing, yielding faster empirical runtimes. It would be good to see some empirical comparison to this family of SMC algorithms, as they’re a much stronger baseline than GCD or ASAp. For example, even ignoring parallelism speedups, MCMC restart for k=10 steps vs. SMC with k=10 particles followed by a final resample proportional to the weights should give a compute-matched (FLOPs) way to generate 1 sample. It would be ideal to see these experiments. Minimally, I would like to see this comparison further discussed and this paper more accurately put into the context of related work.
- Very limited evaluation: 1 model and 1 task with 2 domains and weak baselines.

References
- Loula, J., ... & O'Donnell, T. J. (2024). Syntactic and semantic control of large language models via sequential monte carlo. arXiv preprint arXiv:2504.13139.
- Ahmed, K., Chang, K. W., & Broeck, G. V. D. (2024). Controllable Generation via Locally Constrained Resampling. arXiv preprint arXiv:2410.13111.
- Lipkin, B., ... & Vieira, T. (2025). Fast Controlled Generation from Language Models with Adaptive Weighted Rejection Sampling. arXiv preprint arXiv:2504.05410.

---

> ### Author Rebuttal · Authors · 2025-07-31
>
> > The paper neglects a body of highly related work, missing high-performing baselines. Loula et al (2024), Ahmed et al (2024), and Lipkin et al (2025) also presented this same problem of GCD distortion. …  even ignoring parallelism speedups, MCMC restart for k=10 steps vs. SMC with k=10 particles followed by a final resample proportional to the weights should give a compute-matched (FLOPs) way to generate 1 sample. It would be ideal to see these experiments. Minimally, I would like to see this comparison further discussed and this paper more accurately put into the context of related work.
>
> We thank the reviewer for highlighting this highly relevant literature. We acknowledge that these works were not cited in our original submission. Notably, two of the references were either unpublished or had only very recently appeared at the time of submission, which contributed to their omission. We agree with the reviewer that comparison with this concurrent work adds to the quality of our paper.
>
> The approaches presented in [1] are the ones most similar to our proposed method, since both apply Monte Carlo methods over token sequences to approximate $P^G$ ($g$ if using the notation from [1]). Whereas we approach this problem by using MCMC, [1] implements both Resampled Importance Sampling and Sequential Monte Carlo (SMC) or Particle Filtering. Any of these methods are theoretically guaranteed to produce samples from the target distribution when $N$ tends to infinity, but MCMC does not directly reduce to any of the other two algorithms or vice versa.
>
> Before moving to the theoretical comparison of the two approaches, we addressed the reviewer's suggestion and ran a preliminary evaluation comparing our proposed method with the SMC-AWRS method proposed in [1] and [3]. On a subset (due to timing constraints) of 8/14 BV4 program synthesis tasks, we observe that MCMC-restart achieves 1.86x (geo. mean) lower (i.e., better) KL-divergence on compute-matched runs (k=10 steps vs k=10 particles) vs SMC. We will add a detailed, more comprehensive version of this experiment to the paper.
>
> A clear advantage of the alternative methods in [1] is the inclusion of expensive potentials to score the quality of generations. This is done by adjusting the probability of sequences by the (ratio of) change of potential induced by each of its next-token completions. This is a nice way of integrating such signals into the decoding process, and the details on how to add an analogous factor to the Metropolis-Hastings equations could be a direction of further research. However, in the case of simple grammar-constrained decoding which we study, this notion goes unused, and these approaches reduce to the vanilla versions of IS and SMC.
>
> There are further connections that can be drawn between our approach and those presented in [1]: Importance Sampling and MCMC-restart lend themselves to more aggressive parallelization, since one can draw N complete samples independently and perform one final round of drawing to obtain a final sample. On the other hand SMC is more similar to MCMC-uniform or MCMC-restart, in the sense that some additional computation has to be made at each generation step of the sequences, although for MCMC this computation is only dependent on the previous state of the same sequence, whereas for SMC it depends on the state of the rest of the particles as well.
>
> Overall, we find no strong formal argument to position either one with clear advantage over the others in the setting of grammar constrained decoding. We will add a detailed comparison to our paper.
>
> With respect to [3], we find that the goal of this paper is slightly different from ours. Whereas we deal with the problem of approximately sampling from the constrained distribution across sequences of tokens, [3] is concerned about sampling from the constrained distribution of possible next-tokens given a prefix [3] (Eq. 1). The goal is to avoid potentially expensive masking operations over the entire vocabulary of LMs, and replace them with a rejection sampling approach. The main algorithm proposed, AWRS, is evaluated as a subcomponent of the SMC algorithm, where it in fact yields performance improvements. The question then arises of whether AWRS can be integrated into our MCMC approach. We lean toward the positive, and leave further analysis to future work.
>
> Finally, [2] introduces another solution to the locally constrained decoding distortion problem by sampling a constrained sequence in a 3-step fashion: 1) obtain an unconstrained sample via the autoregressive distribution, 2) project into a constrained space from the pseudolikelihood induced by the unconstrained sample 3) apply resampled importance sampling to correct for bias. The idea of starting from an unconstrained (and undistorted) sample and projecting that to the constrained set is appealing as a proposal distribution for the MCMC approach as well. The projection operator introduced in [2] however seems to require a non-trivial implementation and it is not clear whether it is optimal or an alternative could be more effective in practice. This is again a direction worth exploring and integrating in the future.
>
> In summary, the various existing solutions to the problem we explore in our paper, are largely orthogonal and have merits of their own, and their different components are amenable to be integrated directly into our proposed solution, or vice versa, as part of future research in this area.
>
> [1]  Loula, J., ... & O'Donnell, T. J. (2024).
>
> [2]  Ahmed, K., Chang, K. W., & Broeck, G. V. D. (2024).
>
> [3]  Lipkin, B., ... & Vieira, T. (2025).
>
> > Very limited evaluation: 1 model and 1 task with 2 domains and weak baselines.
>
> Due to space limitation in the response, we kindly ask the reviewer to consult the answers to Reviewer 2 for additional experiments involving further models and to Reviewer 1 for a clarification that our evaluation involves more than just one task.
>
> > Can you include some plots/tables of how long it takes to generate the seed samples across the different conditions? Since it presumably takes ~10x longer to produce 100 samples from the MCMC (k=10) condition than GCD, using 100 seed samples from each seems to be an odd comparison as MCMC spends considerably more time refining each sample. A more fair comparison might be to produce however many K samples you can from each condition in some fixed time budget then use all those seeds.
>
> This is a very good point and we thank the reviewer for this question. To answer it we performed two additional experiments:
>
> - **Time taken to generate**: On our hardware (8x NVIDIA RTX 2080 Ti), for the fuzzing application in Section 4.2, producing a single GCD sample takes ~20 seconds. Our MCMC methods with k=10 steps take ~210 seconds per sample (as expected, ~10x longer), as they perform 10 MCMC steps (though the individual samples of the Restart method could be parallelized). We will add detailed tables.
> - **Adapting number of seeds based on time taken to generate**: To answer the second part of the reviewer’s question, we ran the fuzzer using different numbers of seeds. Our experiment (see Tables below) shows that fidelity in sampling is more important than quantity: our MCMC methods with just 50 seeds (sampled with k=10) are able to achieve higher coverage than GCD using 500 seeds (i.e., the seeds sampled using within the same time budget). We believe that having more high-quality seeds decreases the fuzzer performance as it dilutes its mutation budget across multiple seeds (if the seeds are not good, this results in low coverage).
> ### XML branch coverage after 1 hour of fuzzing (Llama-3.1-8B, varying number of seeds N):
> | **Method**               | **N=50**   | **N=100**  | **N=200**  | **N=500**  |
> |--------------------------|------------|------------|------------|------------|
> | GCD                      | 10.65   | 10.67   | 10.76   | 10.88   |
> | Grammarinator            | 8.41    | 8.49   | 8.51    | 8.53  |
> | MCMC-Uniform-10          | 12.10   | 12.16   | 12.38   | 12.47   |
> | MCMC-Priority-10         | 12.21   | 12.81   | 12.86   | 12.90   |
> | MCMC-Restart-10          | 11.69   | 12.00   | 12.02   | 12.15   |
> ### SQL branch coverage after 1 hour of fuzzing (Llama-3.1-8B, varying number of seeds N):
> | **Method**                | **N=50**    | **N=100**   | **N=200**   | **N=500**   |
> |--------------------------|-------------|-------------|-------------|-------------|
> | GCD                      | 27.27  | 28.26  | 28.46    | 28.39  |
> | Grammarinator            | 23.48  | 25.04   | 26.08  | 26.01    |
> | MCMC-Uniform-10          | 30.72   | 32.93  | 32.94   | 33.09    |
> | MCMC-Priority-10         | 31.08   | 33.70   | 33.80    | 33.83    |
> | MCMC-Restart-10          | 30.57   | 32.36  | 32.84   | 32.89    |
>
>
> ### XML branch coverage after 1 hour of fuzzing (Qwen-2.5-7B, varying number of seeds N):
>
> | **Method**                | **N=50**   | **N=100**  | **N=200**  | **N=500**  |
> |--------------------------|------------|------------|------------|------------|
> | GCD                      | 10.76  | 10.80   | 10.81   | 10.78  |
> | Grammarinator            | 8.41  | 8.49   | 8.51    | 8.53  |
> | MCMC-Uniform-10          | 11.98   | 12.02   | 12.18   | 12.27  |
> | MCMC-Priority-10         | 12.02   | 12.09   | 12.21  | 12.25  |
> | MCMC-Restart-10          | 11.76   | 11.80   | 11.90   | 12.06  |
>
> ### SQL branch coverage after 1 hour of fuzzing (Qwen-2.5-7B, varying number of seeds N):
>
> | **Method**       | **N=50** | **N=100** | **N=200** | **N=500** |
> | ---------------- | -------- | --------- | --------- | --------- |
> | GCD              | 26.32    | 28.15     | 28.37     | 28.15     |
> | Grammarinator    | 23.48    | 25.04     | 26.08     | 26.01     |
> | MCMC-Uniform-10  | 30.15    | 32.20     | 32.32     | 32.36     |
> | MCMC-Priority-10 | 31.05    | 32.75     | 32.97     | 33.00     |
> | MCMC-Restart-10  | 30.44    | 31.85     | 31.85     | 32.02     |

---

> > ### Comment · Reviewer_4jfZ · 2025-08-01
> > **Thank you for additional experiments and discussion of related work**
> >
> > Thank you for the additional experiments and the expanded discussion of related work. I find these new elements add important and critical context to this paper. With their addition, I recommend acceptance, and have updated my score. This is an important topic area with a clear problem presentation and sound approach.

---

### Official Review · Reviewer_NeNk · 2025-07-01

**Clarity:** 3
**Significance:** 2
**Originality:** 2
**Rating:** 5
**Confidence:** 3

**Summary:**

The manuscript proposes a grammar-aligned Markov-Chain Monte-Carlo (MCMC) framework that turns any grammar-constrained decoder (GCD) into a provably correct sampler for language-model outputs under hard constraints. The authors apply a Metropolis–Hastings acceptance test based on the language model likelihood. The framework ensures three key properties of the generated chain:
- It never violates the grammar.
- It converges monotonically to the true conditional distribution.
- In practice, it yields high-quality samples after only a few rejection steps.

Experiments on the SyGuS synthesis benchmark, constituency parsing, and real-world fuzzing of libxml2 and SQLite show large reductions in empirical KL-divergence compared to GCD or ASAp.
The authors investigate three concrete proposal distributions: Restart, Uniform, and Priority, which offer different speed/quality trade-offs for constrained sampling.

**Questions:**

1. Beyond CFGs: How would the Metropolis–Hastings acceptance behave for grammars that are not context-free?

**Ethical Concerns:**

["NO or VERY MINOR ethics concerns only"]

**Final Justification:**

The additional ablations performed by the authors strengthen the paper and make it an important work for constraint decoding. Therefor,e I have updated my score to recommend acceptance of the paper.

**Limitations:**

yes

**Paper Formatting Concerns:**

No issues

**Quality:**

2

**Strengths And Weaknesses:**

## Weaknesses
- Limited model diversity in experiments: all experiments use only Llama-3.1-8B-Instruct. It is unsure how the behavior of constrained sampling may vart from a model to another, especially larger models.

- Narrow experimental scope: the authors experiment with just two fuzzing targets and use single-hour runs. Other targets/domains remain untested (e.g., dialogue).

- Restriction to CFGs: it seems the method does not extend to context-sensitive or semantic constraints enforced by validators.

## Strengths:
- Clear theoretical grounding: the paper formalizes three desiderata (constraint-satisfying, monotonically converging, efficient) and proves the proposed algorithm meets the first two.

- Simplicity and practicality: wraps any off-the-shelf GCD. There is no need for retraining or surrogate models.

- Consistent empirical evidence: lower KL on three benchmark suites and up to 1.2 × branch-coverage gains in fuzzing compared with prior samplers.

---

> ### Author Rebuttal · Authors · 2025-07-31
>
> We thank the reviewer for their detailed and thoughtful evaluation of our work. We appreciate the recognition of the theoretical grounding, simplicity, and empirical effectiveness of our proposed framework. We also value the constructive comments regarding model diversity, experimental scope, and the limitations to context-free grammars, which we will address in the revised version.
>
>
> > Limited model diversity in experiments: all experiments use only Llama-3.1-8B-Instruct. … especially larger models.
>
> We agree with the reviewer’s assessment and have run additional evaluation for a subset of our benchmark tasks during the response period, and will include comprehensive runs of these experiments in the final version. We emphasize that no modifications to our implementation are needed to switch to different models and their specific tokenization schemes..
>
> For the program synthesis benchmarks, we have run the BV4 benchmarks with 3 additional SOTA models: `Qwen2.5-Coder-7B-Instruct`, `gemma-2-9b-it`, and `deepseek-coder-7b-instruct-v1.5`. Across all 14 tasks and 3 models, we see that samples from each of the MCMC instances have lower KL-divergence w.r.t $P^G$ than GCD, with improvements rates presented in the following table
>
> |  | Uniform | Priority | Restart |
> |--|--|--|--|
> | Qwen2.5-Coder-7B-Instruct | 1.23x | 1.42x | 2.71x |
> | gemma-2-9b-it | 1.78x | 2.11x | 3.83x |
> | deepseek-coder-7b-instruct-v1.5 | 2.86x | 3.45x | 4.27x |
>
> These numbers align with the results we originally reported in the main text. Generally, the convergence curves for the new models strongly resemble those from our original evaluation, with fast convergence at the first few steps, tapering off closer to k=10. Also, MCMC-restart continues to display faster convergence than the other 2 proposals, although the final KL-divergence value at k=10 is similar for all flavors.
>
> We also ran the fuzzing tasks from Section 4.2, using Qwen2.5-Coder-7B-Instruct (due to the limited time and resources, we could not run more models on this benchmark). The detailed results are shown in the tables below, but in summary the results for this new model are consistent with our reported findings: MCMC-based methods again demonstrate similar KL-divergence trends and a sustained advantage in coverage over the GCD and Grammarinator baselines. The relative performance ordering of all methods remained the same across both language models.
> ### XML KL-divergence for different MCMC-methods (Qwen-2.5-7B, step count k):
> | **Method**  | **k=2** | **k=3** | **k=4** | **k=5** | **k=6** | **k=7** | **k=8** | **k=9** | **k=10** |
> | ------------- | ---------- | ---------- | ---------- | ---------- | ---------- | ---------- | ---------- | ---------- | ----------- |
> | MCMC-Uniform  | 167   | 149    | 148     | 140   | 133   | 131    | 126      | 133      | 128         |
> | MCMC-Priority    | 126    | 113      | 102  | 101     | 104   | 99  | 100    | 96         | 94          |
> | MCMC-Restart   | 75     | 66        | 67    | 73       | 74      | 72      | 69    | 71    | 77          |
>
> ### SQL KL-divergence for different MCMC-methods (Qwen-2.5-7B, step count k):
>
> | **Method**   | **k=2** | **k=3** | **k=4** | **k=5** | **k=6** | **k=7** | **k=8** | **k=9** | **k=10** |
> | ------------- | ---------- | ---------- | ---------- | ---------- | ---------- | ---------- | ---------- | ---------- | ----------- |
> | MCMC-Uniform   | 134      | 117      | 107      | 104      | 99       | 90       | 89     | 87  | 78        |
> | MCMC-Priority   | 77       | 78       | 75       | 72       | 68       | 59       | 55       | 58       | 68        |
> | MCMC-Restart   | 76       | 70       | 47       | 44       | 41       | 41       | 45       | 46       | 47        |
>
> ### XML branch coverage after 1 hour of fuzzing:
> | **Method**       | **Llama-3.1-8B** | **Qwen-2.5-7B** |
> | ---------------- | ---------------- | --------------- |
> | GCD              | 10.67            | 10.76           |
> | Grammarinator    | 8.49             | 8.41            |
> | MCMC-Uniform-10  | 12.16            | 11.98           |
> | MCMC-Priority-10 | 12.81            | 12.02           |
> | MCMC-Restart-10  | 12.00            | 11.76           |
>
> ### SQL branch coverage after 1 hour of fuzzing:
>
> | **Method**       | **Llama-3.1-8B** | **Qwen-2.5-7B** |
> | ---------------- | ---------------- | --------------- |
> | GCD              | 28.26            | 26.32           |
> | Grammarinator    | 25.04            | 23.48           |
> | MCMC-Uniform-10  | 32.93            | 30.15           |
> | MCMC-Priority-10 | 33.70            | 31.05           |
> | MCMC-Restart-10  | 32.36            | 30.44           |
>
>
>
>
> > … just two fuzzing targets and use single-hour runs. Other targets/domains remain untested (e.g., dialogue).
>
> Thanks for the observation. To evaluate the long-term impact of the generated seeds, we have conducted new, longer fuzzing experiments for up to 6 hours (see tables below).
>
> In the 6-hour runs, we see that for the XML target, coverage gains are minimal after the first hour. However, for the SQL target we observe sustained coverage growth over the 6 hours. Remarkably, the performance advantage of the MCMC methods widens over time.
> We will gladly include this longer running experiment (as well as a 24-hr one in the paper).
>
> ### XML branch coverage for corpus of 100 seeds (Llama-3.1-8B, varying fuzzing duration T):
>
>
> | **Method**       | **T=1 h** | **T=2 h** | **T=3 h** | **T=6 h** |
> | ---------------- | --------- | --------- | --------- | --------- |
> | GCD              | 10.67     | 10.71     | 10.72     | 10.73     |
> | Grammarinator    | 8.49      | 8.56      | 8.59      | 8.63      |
> | MCMC-Uniform-10  | 12.16     | 12.19     | 12.20     | 12.21     |
> | MCMC-Priority-10 | 12.81     | 12.82     | 12.84     | 12.85     |
> | MCMC-Restart-10  | 12.00     | 12.06     | 12.07     | 12.09     |
>
> ### SQL branch coverage for corpus of 100 seeds (Llama-3.1-8B, varying fuzzing duration T):
>
> | **Method**       | **T=1 h** | **T=2 h** | **T=3 h** | **T=6 h** |
> | ---------------- | --------- | --------- | --------- | --------- |
> | GCD              | 28.26     | 29.84     | 31.04     | 32.49     |
> | Grammarinator    | 25.04     | 26.15     | 27.84     | 28.96     |
> | MCMC-Uniform-10  | 32.93     | 33.91     | 35.62     | 37.23     |
> | MCMC-Priority-10 | 33.70     | 35.12     | 37.43     | 39.85     |
> | MCMC-Restart-10  | 32.36     | 33.54     | 34.78     | 36.12     |
>
> We were not able to run a completely new application within the time of the rebuttal, but we will gladly add one more domain in the final version.
>
>
>
>
> > Restriction to CFGs: it seems the method does not extend to context-sensitive or semantic constraints enforced by validators. Beyond CFGs: How would the Metropolis–Hastings acceptance behave for grammars that are not context-free?
>
> This is an excellent point, and we are happy to elaborate further on it. While our method is presented in the context of grammar-constrained decoding, it is important to note that it is not limited to it, and in fact it is generally applicable whenever the constraint is checkable in an incremental way for sequence prefixes. For instance, [1] presents a benchmark of context-sensitive pattern matching tasks, where an incremental pattern validator is available. Another example is Type-Constrained Code Generation [2], an exciting application where constraints are also computed in an incremental manner. For both of these constrained decoders, our approach could be applied.
>
> [1] Lipkin, B., ... & Vieira, T. (2025). Fast Controlled Generation from Language Models with Adaptive Weighted Rejection Sampling. arXiv preprint arXiv:2504.05410.
>
> [2] Niels Mündler, Jingxuan He, Hao Wang, Koushik Sen, Dawn Song, and Martin Vechev.. Type-Constrained Code Generation with Language Models. PLDI 2025

---

> > ### Comment · Reviewer_NeNk · 2025-08-05
> >
> > I thank the authors for addressing all my concerns and presenting new empirical evidence. I believe these additional ablations strengthen the manuscript, and I recommend appending them to the paper.
> >
> > With these additions, I recommend acceptance of the paper, showing an important perspective on constrained decoding.

---

### Official Review · Reviewer_gYBz · 2025-07-03

**Clarity:** 4
**Significance:** 3
**Originality:** 3
**Rating:** 5
**Confidence:** 3

**Summary:**

This paper presents a constrainted sampling setup inspired by the MCMC framework. The proposed approach aims to address some of the challenges faced in current constrainted sampling setups where classic approaches can be inefficient and do not guarantee to recover the original distribution. The proposed approach iteratively refines the generated sample while uses a grammar-constratined decoding setup. The paper evalutes the proposed approach to show certain properties are presevered and further experiments on a practial application of fuzzing to demonstrate practical impact of the approach.

**Questions:**

- How does different tokenization setups of the different LLMs affect the approach? For example, a different LLM may tokenize a particular piece or terminal of CFG differently, does the proposed approach require much changes to be able to handle that?
- What is the initial coverage of the 100-seed problems generated by each approach? How does the coverage change as we increase the number of seeds to 1000 programs?

**Ethical Concerns:**

["NO or VERY MINOR ethics concerns only"]

**Final Justification:**

In my opinion the motivation of this work with regards to the prior approaches and what their limitations are: inefficiency in rejection sampling and disrupting the underlying distribution when doing classic constrained decoding is quite clear and easily understandable. I also believe that the authors proposed an easy and intutive approach which works. Furthermore the authors have also added additional experimental results which answered some of my concerns. As such I remain positive regarding this work.

**Limitations:**

yes

**Paper Formatting Concerns:**

No paper formatting concerns

**Quality:**

3

**Strengths And Weaknesses:**

## strengths
- To the best of my knowledge the proposed approach to perform constrained decoding using MCMC algorithm has not been previously done
- The motivation of this work with regards to the prior approaches and what their limitations are: inefficiency in rejection sampling and disrupting the underlying distribution when doing classic constrained decoding is extremely clear and presented in a nice way
- The idea presented in the paper is simple and intuitive to understand
- The additional practical evaluation in terms of fuzzing to showcase the practical impact of adopting the proposed technique is well-appreciated

## weaknesses
- The main weakness of the work is the limited evaluation in terms of the model used. The work only really uses a single model, it's unclear why this model is chosen or if the technique will work different or require additional work when we switch to a different model (say with a different tokenization scheme)
- Furthermore, "out of 500 samples produced by Llama-3.1-8B-Instruct,
98 only 2 (0.4%) satisfied the constraint imposed by the grammar." This point raised by the author I would guess is only true because of the small size of the model. A larger sota model would like produce many samples with valid grammar (hence the increase in popularity of rejection sampling)
- For the fuzzing experiments, it's unclear why the authors only evaluated 100 seed settings. It would be much more interesting to show how the coverage changes as we increase the number of seeds (i.e., how much "good" coverage can be produced in the seed programs). This would show how the technique gives better diversity

Nevertheless, these are small limitations which should be simple and easy for the authors to address in the incoming revisions

### minor

- line 47 has additional period in the middle of sentence

---

> ### Author Rebuttal · Authors · 2025-07-31
>
> We thank the reviewer for their thorough and constructive feedback. We are pleased that the novelty, clarity, and practicality of our approach—particularly its application to fuzzing—were well recognized. We also appreciate the thoughtful questions and suggestions regarding model choice, evaluation scale, and the impact of tokenization, which we will carefully address in our revision.
>
>
> > only really uses a single model…How does different tokenization setups of the different LLMs affect the approach?
>
> We agree with the reviewer’s assessment and have run additional evaluation for a subset of our benchmark tasks during the response period, and will include comprehensive runs of these experiments in the final version. We emphasize that no modifications to our implementation are needed to switch to different models and their specific tokenization schemes..
>
> For the program synthesis benchmarks, we have run the BV4 benchmarks with 3 additional SOTA models: `Qwen2.5-Coder-7B-Instruct`, `gemma-2-9b-it`, and `deepseek-coder-7b-instruct-v1.5`. Across all 14 tasks and 3 models, we see that samples from each of the MCMC instances have lower KL-divergence w.r.t  $P^G$  than GCD, with improvements rates presented in the following table
>
> |  | Uniform | Priority | Restart |
> |--|--|--|--|
> | Qwen2.5-Coder-7B-Instruct | 1.23x | 1.42x | 2.71x |
> | gemma-2-9b-it | 1.78x | 2.11x | 3.83x |
> | deepseek-coder-7b-instruct-v1.5 | 2.86x | 3.45x | 4.27x |
>
> These numbers align with the results we originally reported in the main text. Generally, the convergence curves for the new models strongly resemble those from our original evaluation, with fast convergence at the first few steps, tapering off closer to k=10. Also, MCMC-restart continues to display faster convergence than the other 2 proposals, although the final KL-divergence value at k=10 is similar for all flavors.
>
> We also ran the fuzzing tasks from Section 4.2, using Qwen2.5-Coder-7B-Instruct (due to the limited time and resources, we could not run more models on this benchmark). The detailed results are shown in the tables below, but in summary the results for this new model are consistent with our reported findings: MCMC-based methods again demonstrate similar KL-divergence trends and a sustained advantage in coverage over the GCD and Grammarinator baselines. The relative performance ordering of all methods remained the same across both language models.
> ### XML KL-divergence for different MCMC-methods (Qwen-2.5-7B, step count k):
> | **Method**  | **k=2** | **k=3** | **k=4** | **k=5** | **k=6** | **k=7** | **k=8** | **k=9** | **k=10** |
> | ------------- | ---------- | ---------- | ---------- | ---------- | ---------- | ---------- | ---------- | ---------- | ----------- |
> | MCMC-Uniform  | 167   | 149    | 148     | 140   | 133   | 131    | 126      | 133      | 128         |
> | MCMC-Priority    | 126    | 113      | 102  | 101     | 104   | 99  | 100    | 96         | 94          |
> | MCMC-Restart   | 75     | 66        | 67    | 73       | 74      | 72      | 69    | 71    | 77          |
>
> ### SQL KL-divergence for different MCMC-methods (Qwen-2.5-7B, step count k):
>
> | **Method**   | **k=2** | **k=3** | **k=4** | **k=5** | **k=6** | **k=7** | **k=8** | **k=9** | **k=10** |
> | ------------- | ---------- | ---------- | ---------- | ---------- | ---------- | ---------- | ---------- | ---------- | ----------- |
> | MCMC-Uniform   | 134      | 117      | 107      | 104      | 99       | 90       | 89     | 87  | 78        |
> | MCMC-Priority   | 77       | 78       | 75       | 72       | 68       | 59       | 55       | 58       | 68        |
> | MCMC-Restart   | 76       | 70       | 47       | 44       | 41       | 41       | 45       | 46       | 47        |
>
> ### XML branch coverage after 1 hour of fuzzing:
> | **Method**       | **Llama-3.1-8B** | **Qwen-2.5-7B** |
> | ---------------- | ---------------- | --------------- |
> | GCD              | 10.67            | 10.76           |
> | Grammarinator    | 8.49             | 8.41            |
> | MCMC-Uniform-10  | 12.16            | 11.98           |
> | MCMC-Priority-10 | 12.81            | 12.02           |
> | MCMC-Restart-10  | 12.00            | 11.76           |
>
> ### SQL branch coverage after 1 hour of fuzzing:
>
> | **Method**       | **Llama-3.1-8B** | **Qwen-2.5-7B** |
> | ---------------- | ---------------- | --------------- |
> | GCD              | 28.26            | 26.32           |
> | Grammarinator    | 25.04            | 23.48           |
> | MCMC-Uniform-10  | 32.93            | 30.15           |
> | MCMC-Priority-10 | 33.70            | 31.05           |
> | MCMC-Restart-10  | 32.36            | 30.44           |
>
>
>
> > A larger sota model would likely produce many samples with valid grammar (hence the increase in popularity of rejection sampling)
>
> To address the reviewer's question, we performed a new rejection-sampling experiment with Qwen2.5-Coder-7B-Instruct and a larger Qwen2.5-Coder-32B-Instruct model. The results are consistent with our original findings: the success rates remained low, at 0.6% (3/500 valid samples) for the 32B model, and 0.2% (1/500 valid samples) for the 7B model.
> We agree with the reviewer that for simple or cleverly prompted constraints, larger models can improve the effectiveness of rejection sampling.
>
> > How does the coverage change as we increase the number of seeds?
>
> We embraced the reviewer’s suggestion and conducted a new set of experiments extending our original methodology. For each generation method, we ran 1-hr fuzzing campaigns with varying numbers of initial seeds: 50, 100, 200, and 500 (we show results for Llama-3.1-8B as well as one of the additional models, Qwen2.5-Coder-7B-Instruct, used in the rest of the author response to show that our results are not specific to one model). (We can include an experiment with 1000 seeds for the final version of the paper as such an experiment requires more time than we had available). This experiment directly measures how the diversity and quality of the initial seed pool impact the fuzzer's ability to discover new coverage over a sustained run.
>
> The tables below show:
> - Adding more seeds generally improves coverage for all methods
> - Our MCMC-based methods outperform the baselines (GCD and Grammarinator) at every seed count. Particularly, even at the lowest seed count (50) the MCMC methods all achieve a higher coverage than the GCD baseline with the highest seed count (500). We will gladly add this experiment to the paper to show the importance of diversity in samples (rather than quantity).
> ### XML branch coverage after 1 hour of fuzzing (Llama-3.1-8B, varying number of seeds N):
> | **Method**               | **N=50**   | **N=100**  | **N=200**  | **N=500**  |
> |--------------------------|------------|------------|------------|------------|
> | GCD                      | 10.65   | 10.67   | 10.76   | 10.88   |
> | Grammarinator            | 8.41    | 8.49   | 8.51    | 8.53  |
> | MCMC-Uniform-10          | 12.10   | 12.16   | 12.38   | 12.47   |
> | MCMC-Priority-10         | 12.21   | 12.81   | 12.86   | 12.90   |
> | MCMC-Restart-10          | 11.69   | 12.00   | 12.02   | 12.15   |
> ### SQL branch coverage after 1 hour of fuzzing (Llama-3.1-8B, varying number of seeds N):
> | **Method**                | **N=50**    | **N=100**   | **N=200**   | **N=500**   |
> |--------------------------|-------------|-------------|-------------|-------------|
> | GCD                      | 27.27  | 28.26  | 28.46    | 28.39  |
> | Grammarinator            | 23.48  | 25.04   | 26.08  | 26.01    |
> | MCMC-Uniform-10          | 30.72   | 32.93  | 32.94   | 33.09    |
> | MCMC-Priority-10         | 31.08   | 33.70   | 33.80    | 33.83    |
> | MCMC-Restart-10          | 30.57   | 32.36  | 32.84   | 32.89    |
>
>
> ### XML branch coverage after 1 hour of fuzzing (Qwen-2.5-7B, varying number of seeds N):
>
> | **Method**                | **N=50**   | **N=100**  | **N=200**  | **N=500**  |
> |--------------------------|------------|------------|------------|------------|
> | GCD                      | 10.76  | 10.80   | 10.81   | 10.78  |
> | Grammarinator            | 8.41  | 8.49   | 8.51    | 8.53  |
> | MCMC-Uniform-10          | 11.98   | 12.02   | 12.18   | 12.27  |
> | MCMC-Priority-10         | 12.02   | 12.09   | 12.21  | 12.25  |
> | MCMC-Restart-10          | 11.76   | 11.80   | 11.90   | 12.06  |
>
> ### SQL branch coverage after 1 hour of fuzzing (Qwen-2.5-7B, varying number of seeds N):
>
> | **Method**       | **N=50** | **N=100** | **N=200** | **N=500** |
> | ---------------- | -------- | --------- | --------- | --------- |
> | GCD              | 26.32    | 28.15     | 28.37     | 28.15     |
> | Grammarinator    | 23.48    | 25.04     | 26.08     | 26.01     |
> | MCMC-Uniform-10  | 30.15    | 32.20     | 32.32     | 32.36     |
> | MCMC-Priority-10 | 31.05    | 32.75     | 32.97     | 33.00     |
> | MCMC-Restart-10  | 30.44    | 31.85     | 31.85     | 32.02     |
>
>
>
> > How does different tokenization setups of the different LLMs affect the approach? For example, a different LLM may tokenize a particular piece or terminal of CFG differently, does the proposed approach require much changes to be able to handle that?
>
> Our implementation builds on existing grammar-constrained decoding (GCD) implementations and does not require any changes when changing the model, as long as the model is supported by the underlying GCD library. That is, the problem of dealing with how different tokens match different terminals in the grammar is handled by the GCD implementation and our sampling approach does not have to worry about it.

---

> > ### Comment · Reviewer_gYBz · 2025-08-04
> >
> > Thanks to the authors for the response to all my questions. In particular thanks for running additional experiments including the ones regarding different models. Although I still think that running ~7b models are not that representative I do appreciate the additional models that the authors have tested on. Furthermore I think the new coverage results by adding more seeds as an experimental setting is also encourage. As such, I remain positive on the work!

---

### Official Review · Reviewer_Q6rk · 2025-07-03

**Clarity:** 3
**Significance:** 3
**Originality:** 2
**Rating:** 4
**Confidence:** 4

**Summary:**

This paper presents a Monte Carlo method for grammar-constrained sampling from language models.
Vanilla grammar-constrained decoding (GCD) results in a skewed distribution that is far from the target constraint distribution.
To solve this problem, this paper proposed a conceptually simple method (a slightly modified Metropolis-Hastings algorithm that allows decoding a new sample from the prefix generated in the previous step rather than from scratch) and showed it to be effective on two tasks.

**Questions:**

Please also refer to the Strengths and Weaknesses part:
* When does the proposed sampling method perform better than a simple resampled importance sampling?

**Ethical Concerns:**

["NO or VERY MINOR ethics concerns only"]

**Final Justification:**

My concerns regarding the proposed method vs. 'restart' are mostly resolved. Thus, I raise my score.

**Limitations:**

Yes

**Quality:**

3

**Strengths And Weaknesses:**

Strengths:
* The paper is well-written and easy to follow.
* The proposed method is simple and effective.


Weaknesses:
My major concern is about the results presented in Fig. 2 (a) and Fig. 3 (a):
From the KL divergence curves, it seems the 'restart' achieves the fastest convergence compared to more sophisticated choices of $p_{pos}^w$, which means a simple resampling importance sampling is sufficient for the task. In that case, how does the extra flexibility in proposal distribution help?

Minor points:
* Experimental results are only reported on two tasks.
* The paper lacks of discussion of and comparison with other Monte Carlo methods, e.g., https://arxiv.org/pdf/2504.13139

---

> ### Author Rebuttal · Authors · 2025-07-31
>
> We thank the reviewer for their thoughtful and constructive feedback. We appreciate the positive comments on the clarity of the writing and the simplicity and effectiveness of our proposed method.
>
> > My major concern is … it seems the 'restart' achieves the fastest convergence … how does the extra flexibility in proposal distribution help? When does the proposed sampling method perform better than a simple resampled importance sampling?
>
> Thanks for the observation. It is true that MCMC-restart generally achieves faster convergence in practice than MCMC-uniform and MCMC-priority, although the difference in convergence speed and final KL-divergence value between the three approaches is not drastic. The main advantage of subsequence resampling proposals, and in particular of MCMC-priority, comes from the “local” diversity they induce in samples—e.g., they produce more variants of one program rather than more different programs. This local diversity is beneficial in the fuzzing task (see Figure 3 in the paper), where although MCMC-restart converges a little faster than MCMC-priority, the samples produced by MCMC-priority result in higher coverage when actually running the fuzzer (i.e., MCMC-priority has better downstream performance).
>
> The motivation for MCMC-uniform and MCMC-priority comes from similar analogous approaches for proposal design that have been successful in stochastic program synthesis [1,2,3]. These instances rely on proposals that locally perturb a substructure of the current candidate. We will clarify and expand on these points in the final version.
>
> As for comparison with resampled importance sampling, we ran a preliminary evaluation comparing our proposed method with the SMC method proposed in [4], which itself is reported to be stronger than Resampled Importance Sampling. On a subset of the BV4 program synthesis tasks, we observe that MCMC-restart achieves 1.86x (geo. mean) better KL-divergence on compute-matched runs (k=10 steps vs k=10 particles) vs SMC. We will add a detailed version of this experiment to the paper.
>
> [1] Eric Schkufza, Rahul Sharma, and Alex Aiken. 2013. Stochastic superoptimization. ASPLOS '13
>
> [2] Aditya V. Nori, Sherjil Ozair, Sriram K. Rajamani, and Deepak Vijaykeerthy. 2015. Efficient synthesis of probabilistic programs. PLDI '15
>
> [3] Feras A. Saad, Marco F. Cusumano-Towner, Ulrich Schaechtle, Martin C. Rinard, and Vikash K. Mansinghka. Bayesian Synthesis of Probabilistic Programs for Automatic Data Modeling. POPL 2019
>
> [4] Loula, J., ... & O'Donnell, T. J. (2024). Syntactic and semantic control of large language models via sequential monte carlo. arXiv preprint arXiv:2504.13139.
>
>
>
>
>
>
>
> > Experimental results are only reported on two tasks
>
> We are grateful for the reviewer’s observation and the opportunity to clarify this point. We consider 2 domains: program synthesis and program fuzzing, with 35 problems for synthesis across 3 splits of tasks (15 for SLIA, 14 for BV4, and 6 for CP) and 2 problems for fuzzing (XML and SQL).
>
> Our rationale when choosing the benchmarks was as follows. First, we chose the three types of program synthesis benchmarks as they are the main evaluation component of the previous work we evaluated against [1]. Second, we designed the program fuzzing evaluation since we were not aware of any task where the quality of the full empirical distribution is paramount; other GCD tasks are mostly concerned with finding one solution rather than exploiting the diversity of the distribution. To our knowledge, our fuzzing task is the first to explicitly evaluate example diversity in this way and we consider the design of this task itself a contribution to the paper.
> We can incorporate relevant domains proposed in the suggested related work for the final version of this work.
>
> [1] Kanghee Park, Jiayu Wang, Taylor Berg-Kirkpatrick, Nadia Polikarpova, and Loris D'Antoni. Grammar Aligned Decoding. NeurIPS 2024
>
>
> > The paper lacks of discussion of and comparison with other Monte Carlo methods, e.g., https://arxiv.org/pdf/2504.13139
>
> Both our approach and the referenced one [1] apply Monte Carlo methods over token sequences to approximate $P^G$ ($g$ if using the notation from [1]). Whereas we approach this problem by using MCMC, [1] implements both Resampled Importance Sampling and Sequential Monte Carlo (SMC) or Particle Filtering. Any of these methods are theoretically guaranteed to produce samples from the target distribution when $N$ tends to infinity, but MCMC does not directly reduce to any of the other two algorithms or vice versa.
>
> Before moving to the theoretical comparison of the two approaches, we ran a preliminary evaluation comparing our proposed method with the SMC method proposed in [1]. On a subset (due to timing constraints) of 8/14 BV4 program synthesis tasks, we observe that MCMC-restart achieves 1.86x (geo. mean) lower (i.e., better) KL-divergence on compute-matched runs (k=10 steps vs k=10 particles) vs SMC. We will add a detailed, more comprehensive version of this experiment to the paper.
>
> A clear advantage of the alternative methods in [1] is the inclusion of expensive potentials to score the quality of generations. This is done by adjusting the probability of sequences by the (ratio of) change of potential induced by each of its next-token completions. This is a nice way of integrating such signals into the decoding process, and the details on how to add an analogous factor to the Metropolis-Hastings equations could be a direction of further research. However, in the case of simple grammar-constrained decoding which we study, this notion goes unused, and these approaches reduce to the vanilla versions of IS and SMC.
>
> There are further connections that can be drawn between our approach and those presented in [1]: Importance Sampling and MCMC-restart lend themselves to more aggressive parallelization, since one can draw N complete samples independently and perform one final round of drawing to obtain a final sample. On the other hand SMC is more similar to MCMC-uniform or MCMC-restart, in the sense that some additional computation has to be made at each generation step of the sequences, although for MCMC this computation is only dependent on the previous state of the same sequence, whereas for SMC it depends on the state of the rest of the particles as well.
>
> Overall, we find no strong formal argument to position either one with clear advantage over the others in the setting of grammar constrained decoding. We will add a detailed comparison to our paper.
>
> [1]  Loula, J., ... & O'Donnell, T. J. (2024). Syntactic and semantic control of large language models via sequential monte carlo. arXiv preprint arXiv:2504.13139.

---

> > ### Comment · Reviewer_Q6rk · 2025-08-05
> >
> > I thank the authors for their response. My concerns are mostly resolved, and I adjust my score accordingly.

---

### Note · Authors · 2025-08-12

We would like to sincerely thank the reviewers for their thoughtful feedback and for giving us the opportunity to address their questions during the rebuttal. We are grateful that all reviewers recommended acceptance following the discussion phase, and we appreciate their engagement and constructive input throughout the process.

---

### Decision · Program_Chairs · 2025-09-17

**Decision:**

Accept (poster)

**Comment:**

**Contribution:** This is a straightforward method for correcting (debiasing) the naive left-to-right algorithm for constrained sampling from autoregressive LMs.  It's MCMC where a proposal consists of throwing away some suffix and regenerating it.  The simplest and most effective version is to throw away the whole string and regenerate it - the independent Metropolis algorithm.  A couple of other heuristics are tried as well.

**Strengths:** The method is unbiased in the limit (for what that's worth), requires no training, is demonstrated to be practical in a situation where rejection sampling is not, and works well enough that it should be considered as a simple baseline beyond rejection sampling by papers that are presenting more complicated methods such as SMC.  There's an interesting downstream evaluation.

**Weaknesses:** The evaluation is mostly internal among variants of the proposal distribution.  It didn't originally compare with other constrained sampling methods.  In the rebuttal, the authors finally gave a preliminary comparison showing that they beat a strong competitor; but it was only under one condition, not a systematic study varying the target distribution and the amount of computation.

Personally, I would also have liked to see study of [additional proposal distributions](https://openreview.net/forum?id=M1b7IuY6Co&noteId=zIDosfH7ka), more constrained generation settings, and [additional evaluation metrics](https://openreview.net/forum?id=M1b7IuY6Co&noteId=LqDKXB2Vz5).  However, the authors did not have an opportunity to reply to these AC comments.

**Most important reason to accept:** Consensus among reviewers.

**Discussion during the rebuttal period:** This was where the authors compared to other methods from the literature (SMC, as well as rejection sampling), which should have been done in the original submission.  However, this comparison was only a "preliminary evaluation"; they promised to "add a detailed version of this experiment to the paper."

The authors also evaluated on additional LLMs as requested by the reviewers, although this may be less important.

-------
# Comments from AC to reviewers during discussion period

Shared here because they may be useful to authors.

## Comment from AC

I believe the problem is presented as drawing a single sample from the target distribution.

However, the proposed MCMC methods actually draw many samples from the proposal distribution; so do the alternatives considered (rejection sampling, importance sampling, SMC).  Each of the proposed and alternative methods can convert its sequence of samples into a unweighted or weighted ensemble of samples, which is biased but becomes unbiased in the infinite-sample limit.  That is commonly used for Monte Carlo estimation of an expectation under the target distribution.

Such an ensemble is an estimate of the probability distribution and should be evaluated on its divergence from the (estimated) target distribution. This is better than simply evaluating the single-sample case as done in the submission, because
* it reduces the variance of the evaluation metric (line 237)
* it allows the computation of other evaluation metrics, such as total variation distance (TVD) (which can be used to bound the error of Monte Carlo expectation estimation)
* the evaluation in the paper favors the MCMC methods because it uses only the last sample from the Markov chain, which is closer in distribution to the target than the earlier "burn-in" samples were.  I believe that with only the last sample, TVD converges faster under independent MH than under resampled importance sampling, but that neither method dominates [the earlier samples are used as well](http://users.stat.umn.edu/~geyer/mcmc/one.html).  So the MCMC methods should be compared to resampled importance sampling (and SMC) in both settings, not just the one that favors the MCMC methods.

Pro tip: For independent Metropolis-Hastings (the paper's MCMC-Restart method), the unweighted ensemble of MCMC states can be converted to a better weighted ensemble by Rao-Blackwellizing the order of the samples ([Atchadé and Perron, 2005](https://www3.stat.sinica.edu.tw/statistica/oldpdf/A15n11.pdf)).

## Comment from AC
Thanks to all of you for your reviews and your discussions with the authors, which I think will improve the paper and help situate it better with respect to other work.

The paper appears to be a good practical nugget - it's nice when simple techniques work.  It seems that you are all recommending acceptance at this point.

I think this algorithm was not entirely unknown in the structured prediction community.  I remember discussing it around 15 years ago :), in the context of Gibbs or MH sampling from an energy-based distribution over the accepting paths of a finite-state automaton (or derivations from a context-free grammar).  However, I can't turn up a mention in the literature.

There are natural generalizations too, including infilling versions that replace an infix rather than a suffix, versions that incorporate the automaton (or grammar) into the proposal distribution, and population-based methods with crossover (evolutionary Monte Carlo).

What do you think of the prioritization heuristic used in this submission (line 202)?  As I read it, it considers the perplexity only at the single next token, which seems surprisingly local and tokenizer-dependent to me.  Also, I wonder why perplexity and not, say, log-perplexity (= entropy).  There is a bunch of prior work on learning where to sample, e.g., https://proceedings.mlr.press/v38/shi15.html , which should be cited, although I suppose this paper is trying to avoid any modeling or learning in the interest of simplicity.

### Reply from Reviewer 4jfZ

Agreed re this problem being known in the structured prediction community for sampling from globally normalized energy-based models. Here are a couple of the earliest works I know that characterized this problem: (1) Lafferty et al (2001) on CRFs and "label bias" in structured prediction (2) Rosenfeld et al (2001) on whole-sentence exponential language models and sampling from these models via MCMC.

That being said, while this issue was better understood by the classical NLP community, it has been mostly ignored as part of the modern LLM discourse, and is worth being restated. There are also unique challenges to this modern instantiation, e.g., alignment mismatch between grammar terminals and LM tokens, very large token vocabulary sizes, etc., that yield different practical tradeoffs with some classical approaches.

Re the priority heuristic, I think it seems reasonable to use these points for refinement. Yes, you are right that this is quite local, i.e., each next-token probability yields the marginal probability over suffixes without the constraint, and after applying the constraint conditional next-token probabilities would change. That being said, doing something more global would require solving an additional inference problem of the same form as the original problem, and this needs to bottom out to a heuristic at some point. The general idea of targeting search at points of "uncertainty" is reasonable at a high-level. I would have found entropy more intuitive, but perplexity is fine. It is probably worth noting, though, that these decisions are relatively arbitrary and that there are plenty of other options one could have considered.

#### Reply from AC
To clarify, I wasn't saying only that the problem was known, but that this particular MCMC sampler was known.
Evaluating it in a modern setting against modern alternatives may still be helpful.